



# Interannual drivers of the seasonal cycle of CO$_2$ fluxes in the Southern Ocean

Luke Gregor[1,2], Schalk Kok[3] and Pedro M. S. Monteiro[1]

[1] Southern Ocean Carbon-Climate Observatory (SOCCO), CSIR, Cape Town, South Africa
[2] University of Cape Town, Department of Oceanography, Cape Town, South Africa
[3] University of Pretoria, Department of Mechanical and Aeronautical Engineering, Pretoria, South Africa

*Correspondence to*: Luke Gregor (luke.gregor@uct.ac.za)

**Abstract.** Machine learning methods (support vector regression and random forest regression) were
used to map gridded estimates of $\Delta p$CO$_2$ in the Southern Ocean from SOCAT v3 data. A low (1° ×
monthly) and high (0.25° × 16-day) resolution implementation of each of these methods as well as the
SOM-FFN method of Landschützer et al. (2014) were added to a five member ensemble. The ensemble
mean $\Delta p$CO$_2$ was used to calculate $F$CO$_2$ (air-sea CO$_2$ flux). Data was separated into nine domains
defined by basin (Indian, Pacific and Atlantic) and biomes defined by Fay and McKinley (2014). The
regional approach showed large zonal asymmetry in $\Delta p$CO$_2$ and $F$CO$_2$ estimates. Importantly, there was
a seasonal decoupling of the modes summer and winter interannual variability. Winter trends had a
larger 10 year mode of variability compared to summer trends, which had a shorter 4–6 year mode. To
understand this variability of $F$CO$_2$, we separately assessed changes in summer and winter $\Delta p$CO$_2$ and
the drivers thereof. The dominant winter changes were driven by wind stress variability. Summer
variability correlated well with chlorophyll-*a* variability where the latter had high concentrations. In
regions of low chlorophyll-*a* concentrations, wind stress and sea surface temperature were lower order
drivers of $\Delta p$CO$_2$.

## 1 Introduction

The Southern Ocean plays a key role in the uptake of anthropogenic CO$_2$ (Khatiwala et al. 2013;
DeVries et al. 2017). Moreover, it has been shown that the Southern Ocean is sensitive to
anthropogenically influenced climate variability, such as the intensification of the westerlies (Le Quéré
et al. 2007; Lenton et al. 2009; Swart and Fyfe 2012; DeVries et al. 2017). Until recently, the
community has not been able to accurately measure the changes, let alone understand the drivers, of
CO$_2$ in the contemporary Southern Ocean due to a paucity of observations (Landschützer et al. 2015).
Empirical models provide an interim solution to this challenge until prognostic ocean biogeochemical
models are able to represent the Southern Ocean CO$_2$ seasonal cycle accurately (Lenton et al. 2013;
Rödenbeck et al. 2015; Mongwe et al. 2016). There is an agreement in the large changes in CO$_2$ fluxes
in the Southern Ocean from a source in the 1990's to a sink in the 2000's; however, there is
disagreement in the drivers of the changes in CO$_2$ uptake (Lovenduski et al. 2008; Landschützer et al.



2015; DeVries et al. 2017). This study aims to understand the drivers of the changing $CO_2$ sink in the Southern Ocean based on an ensemble of empirical estimates with a seasonal analysis framework.

Empirical methods estimate $CO_2$ by extrapolating the sparse ship based $CO_2$ measurements using satellite observable proxies. This approach has allowed for a better understanding of the drivers of $CO_2$
by providing improved spatial and temporal resolution of the variability. Landschützer et al. (2015) used an artificial neural network (ANN) to show that the strengthening of the Southern Ocean $CO_2$ uptake during the period 2000-2010 is part of a decadal internal variability in the natural $CO_2$ flux dynamics. The authors found that the strengthening sink was not due to changes overturning circulation associated with wind stress as suggested in other studies (Lenton and Matear 2007; Lovenduski et al.
2007; Lenton et al. 2009; DeVries et al. 2017). Rather, they suggested that atmospheric circulation has become more zonally asymmetric since the mid 2000's. This led to oceanic dipole of cooling and warming whose net impact together with changes in the DIC/TA was to increase the uptake of $CO_2$ (Landschützer et al. 2015). During this period, in the Atlantic basin, southward advection reduced upwelled DIC in surface waters overcoming the effect of the concomitant warming in the region.
Conversely, in the Eastern Pacific sector of the Southern Ocean, stronger cooling overwhelmed increased upwelling (Landschützer et al. 2015). Munro et al. (2015) supported this mechanism, with data from the Drake Passage showing that $\Delta p CO_2$ decreased between 2002 and 2014.

In a subsequent study Landschützer et al. (2016) proposed that interannual variability of $CO_2$ in the
Southern Ocean is tied to the decadal variability of the Southern Annular Mode (SAM) – the dominant mode of atmospheric variability in the Southern Hemisphere (Marshall 2003). This concurs with previous studies, which suggested that the increase in the SAM during the 1990's resulted in the weakening of the Southern Ocean sink (Le Quéré et al. 2007; Lenton and Matear 2007; Lovenduski et al. 2007; Lenton et al. 2009). The work by Fogt et al. (2012) bridges the gap between the proposed
asymmetric atmospheric circulation of Landschützer et al. (2015) and the observed correlation with the SAM of Landschützer et al. (2016). Their study shows that changes in the SAM have been zonally asymmetric and that that this variability is highly seasonal, thus amplifying or suppressing the amplitude of the seasonal mode.

Assessing the changes through a seasonal framework may thus help shed light on the drivers of $CO_2$ in the Southern Ocean. Southern Ocean seasonal dynamics suggest that the processes driving $pCO_2$ are complex but with two clear contrasting extremes. In winter, the dominant deep mixing and entrainment processes are zonally uniform driving an increase in $pCO_2$ with the region south of the Polar Front (PF) becoming a net source and weakening the net sink north of the PFZ (Lenton et al. 2013). In summer, the
picture is much more spatially heterogeneous, with NPP being the primary driver of variability



(Mahadevan et al. 2011; Thomalla et al. 2011; Lenton et al. 2013). The competition between light and iron limitation results in heterogeneous distribution of Chl-*a* in both space and time, with similar implications for $p$CO$_2$ (Thomalla et al. 2011; Carranza and Gille, 2015). The interaction between the large-scale drivers, such as wind stress, surface heating and mesoscale ocean dynamics, are the primary

cause of this complex picture (McGillicuddy, 2016; Mahadevan et al. 2012). Some regions of elevated mesoscale and submesoscale dynamics, mainly in the Sub-Antarctic Zone (SAZ) are also characterized by strong intraseasonal modes in summer primary production (Thomalla et al. 2011) and $p$CO$_2$ (Monteiro et al. 2015). The magnitudes of these opposing seasonal processes are large, resulting in the seasonal cycle being the dominant mode of variability in the Southern Ocean.


In this study we examine interannual variability in the air-sea fluxes of CO$_2$ between 1998 – 2014 through interannual changes in the characteristics of the seasonal mode of both $p$CO$_2$ and $F$CO$_2$ in the Southern Ocean. We use an ensemble of empirical estimates of CO$_2$ that combine *in situ* observations with remotely sensed proxies to perform this analysis.

## 85  2 Empirical methods and data

### 2.1 Ensemble members

In this study we made use of three empirical methods combined to an ensemble – these methods are presented in Gregor et al. (2017). The advantage of an ensemble over a single method approach is that a degree of robustness is added to the estimate, assuming that ensembles have unique biases in time and

space. The other important assumption we make here is that the majority methods will be correct, while the minority will be biased. The ensemble mean contains five different $\Delta p$CO$_2$ estimate approaches, shown in Table 1, with low- and high-resolution (1°, monthly and 0.25°, 16-day respectively). The SOM-FFN method is defined by Landschützer et al. (2014) and is a two-step neural network approach (trained with SOCAT v2) that first clusters data and then applies a regression model to each cluster

(Bakker et al. 2014). The low resolution implementations of Support Vector Regression (SVR) and Random Forest Regression (RFR) methods are introduced in Gregor et al. (2017). Note that these are trained with SOCAT v3 data (Bakker et al. 2016). The high-resolution implementations of SVR and RFR used in this study are implemented in the same way as described in Gregor et al. (2017). The high-resolution estimates of $\Delta p$CO$_2$ are resampled to match the low resolution data in the ensemble.






**Table 1:** Five empirical methods used in the ensemble. RFR-LR and SVR-LR are described in Gregor et al. (2017). SOM-FFN is from Landschützer et al. **(Landschützer et al. 2014)**. SST = sea surface temperature, MLD = mixed layer depth, SSS = sea surface salinity, ADT = absolute dynamic topography, Chl-$a$ = Chlorophyll-$a$, $p$CO$_{2(atm)}$ = fugacity of atmospheric $CO_2$, $x$CO$_{2(atm)}$ = mole fraction of atmospheric $CO_2$, $\Phi$(lat, lon) = $N$-vector transformations of latitude and longitude, $\lambda$(day of year) = trigonometric transformation of the day of the year.

| Method | Resolution | | Input variables | RMSE |
| --- | --- | --- | --- | --- |
| | Space | Time | | ($\mu$atm) |
| RFR-LR | 1.00° | Month | SST, MLD, SSS, ADT, Chl-$a_{(clim)}$, $p$CO$_{2(atm)}$, $\Phi$(lat, lon), $t$(day of year) | 17.21 |
| SVR-LR | 1.00° | Month | SST, MLD, SSS, ADT, Chl-$a_{(clim)}$, $p$CO$_{2(atm)}$, $\Phi$(lat, lon), $t$(day of year) | 21.73 |
| SOM-FFN | 1.00° | Month | SST, MLD, SSS, Chl-$a$, $x$CO$_{2(atm)}$ | 15.45 |
| RFR-HR | 0.25° | 16-day | SST, MLD, SSS, ADT, Chl-$a_{(clim)}$, $p$CO$_{2(atm)}$ | 12.58 |
| SVR-HR | 0.25° | 16-day | SST, MLD, SSS, ADT, Chl-$a_{(clim)}$, $p$CO$_{2(atm)}$, $\Phi$(lat, lon), $t$(day of year) | 19.18 |

Table 1 also shows the proxy variables used for each of the methods. The sources for the proxy variables are consistent for all methods ensuring a fair comparison. This is particularly important for the assimilated model variables, mixed-layer depth (MLD) and sea surface salinity (SSS) from Estimating the Circulation and Climate of the Ocean, Phase II (ECCO$_2$) (Menemenlis et al. 2008). Choosing to use different products could result in data driven differences (from the same variable). Other data sources are: sea surface temperature (SST) (Reynolds et al. 2007), Chlorophyll-$a$ (Chl-$a$) (Maritorena and Siegel 2005), absolute dynamic topography (ADT) (Duacs), $x$CO$_2$ (CDIAC 2016) with $p$CO$_{2(atm)}$ calculated from interpolated $x$CO$_2$ using NCEP2 sea level pressure (Kanamitsu et al. 2002). Note that ADT coverage is limited to regions of no to very low concentrations of ice cover, thus estimates for SVR and RFR methods do not extend into the ice covered regions during winter.

Seasonality of the data is preserved by transforming the day of the year ($j$) and is included in both SVR and RFR analyses:

$$t = \begin{pmatrix} \cos\left(j \cdot \dfrac{2\pi}{365}\right) \\ \sin\left(j \cdot \dfrac{2\pi}{365}\right) \end{pmatrix} \quad (1)$$

Transformed coordinate vectors were passed to only SVR using $n$-vector transformations of latitude ($\lambda$) and longitude ($\mu$) (Gade 2010; Sasse et al. 2013), with $n$ containing:

$$N = \Phi \begin{pmatrix} \sin(\lambda) \\ \sin(\mu) \cdot \cos(\lambda) \\ -\cos(\mu) \cdot \sin(\lambda) \end{pmatrix} \quad (2)$$

## 2.2 Air-sea CO$_2$ fluxes

Air sea CO$_2$ fluxes are calculated with:

$$F\text{CO}_2 = k_w \cdot K_0 \cdot (p\text{CO}_2^{sea} - p\text{CO}_2^{atm}) \cdot (1 - [ice]) \quad (3)$$



The gas transfer velocity ($k_w$) is calculated using a quadratic dependency of wind speed with the coefficients of (Wanninkhof et al. 2009). Wind speed is calculated from the *u* and *v* vectors $\left(\sqrt{u^2 + v^2}\right)$ of the Cross-Calibrated Multiplatform Product v2 (Atlas et al. 2011). Coefficients from Weiss (1974) are used to calculate $K_0$ and $\Delta p\text{CO}_2$ is estimated by the empirical models. The effect of sea-ice cover on $CO_2$ flux is treated linearly (Butterworth and Miller 2016): the fraction of sea ice cover ([*ice*]) is

converted to fraction of open water by subtracting one as shown in Equation (3).

## 2.3 Regional Coherence Framework

Southern Ocean $CO_2$ is spatially heterogeneous both zonally and meridionally (Jones et al. 2012). In order to understand this heterogeneity we used the three southernmost biomes defined by Fay and McKinley (2017) as done in Rödenbeck et al. (2015). From north to south these are: sub-tropical

seasonally stratified (STSS), sub-polar seasonally stratified (SPSS), seasonally ice covered region (ICE). These three biomes are comparable to the SAZ, PFZ and MIZ respectively and will be used throughout the rest of the study. The Southern Ocean is further split into basins where the boundaries are defined by lines of longitude (70°W : Atlantic : 20°E : Indian : 145°E : Pacific : 70°W).

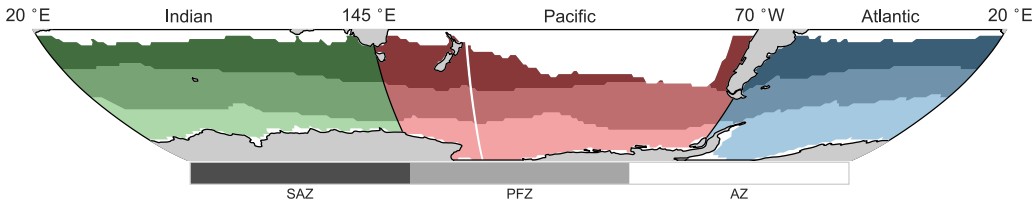


**Figure 1: A map showing the regions used throughout this study. The three biomes, SAZ, PFZ and MIZ used in this study are defined by Fay and McKinley (2014). The regions are also split by basin with boundaries shown above the map.**

## 3 Results and discussion

In this section we present and discuss the data. The first section examines the variability of between the

ensemble members to understand the potential limitations of the dataset. We then look at the seasonal cycle of the ensemble mean in time and space. This is done to lay the foundation knowledge for the interpretation of the results when assessed with the regional framework, which is the following section. In the regional interpretation the data is decomposed into nine regions as shown in Figure 1. The section that follows sets out to make sense of the trends observed in the regional decomposition.

**3.1 Ensemble member performance and variability**

In this section we discuss the performance and variability of the ensemble members. The individual ensemble member scores are shown in Table 1. The RFR-HR is the best performing member, with the lowest RMSE (12.58 μatm). The SVR members score the lowest (21.73 and 19.18 μatm for LR and HR respectively), but were still included due to the method's sensitivity to sparse data, which is favourable





to the poorly sampled winter period (Gregor et al. 2017). This compliments the RFR methods, which

score well (12.58 and 17.21 µatm for HR and LR respectively), but are prone to being insensitive to

sparse data (Gregor et al. 2017). The SOM-FFN member has the best of the low-resolution scores

(15.45 µatm). However, this is because the SOM-FFN is tested with SOCAT v2 data rather than

SOCAT v3, where the latter has a larger standard deviation (32.85 and 36.27 µatm respectively). When

RFR-LR and SVR-LR are tested with the SOCAT v2 dataset, the RMSE scores are 15.15 and 19.82

µatm respectively.

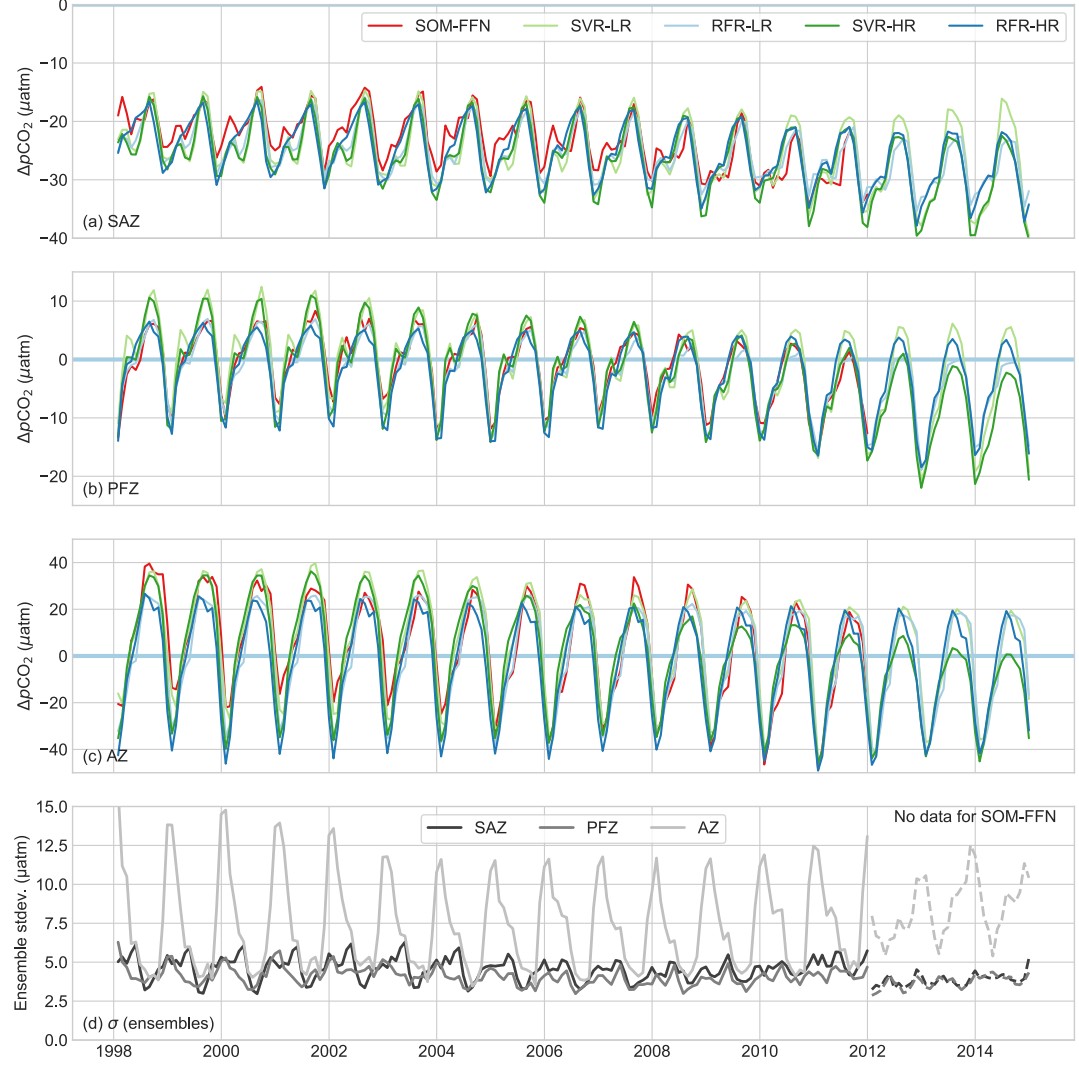

**Figure 2: Time series of the five ensemble members for each biome as defined by Fay and McKinley (2014): (a) SAZ, (b) PFZ, (c) AZ. (d) shows the standard deviation between ensemble members for the three biomes. The SOM-FFN data ends at the end of 2011. This is indicated in (d) by the dashed line.**


Figure 2 shows time series of the each of the methods for the three Southern Ocean biomes as defined

by Fay and McKinley (2014). The methodological and data driven differences between each of the

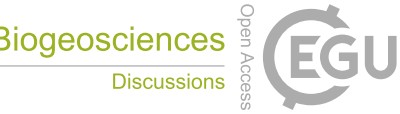


approaches have been addressed in Gregor et al. (2017). In general, there is good agreement amongst
the methods with a few notable exceptions. In the SAZ (Figure 2a) The SOM-FFN differs from all other

methods for summer and autumn from 1998 to 2008. Gregor et al. (2017) attributed this difference to
the clustering step used by the SOM-FFN that created large discrepancies in the Atlantic sector. The
SVR-LR method overestimates $\Delta p\text{CO}_2$ relative to the other methods for winter 2012 to 2014. In the
PFZ (Figure 2b), the SVR methods (LR and HR) overestimate $\Delta p\text{CO}_2$ relative to the other methods
during winter from 1998 to 2004 due to the sensitivity to sparse winter data. The spread of data in the

AZ is much larger than the two other regions, but the impact on the fluxes is reduced due to ice cover
during winter (Ishii et al. 1998; Bakker et al. 2008; Butterworth and Miller 2016).

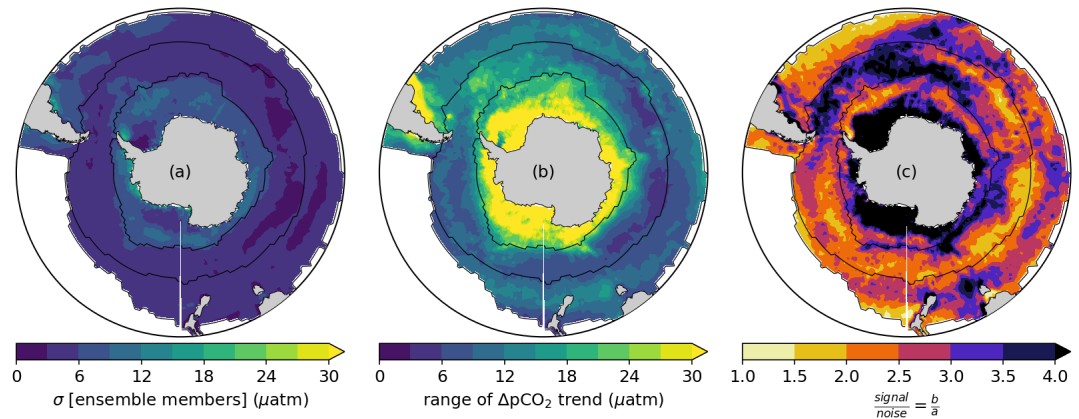

**Figure 3: The mean spatial standard deviation (a) of the ensemble members (SVR-LR, SVR-HR, RFR-LR, RFR-HR and SOM-FFN) is shown to represent the "noise" of the ensemble mean. The "signal" (b) is calculated as the mean difference between the**

**minimum and maximum values of the annually averaged $\Delta p\text{CO}_2$. This shows the signal that needs to be detected by the ensemble. The signal to noise ratio (c) shows regions where the confidence in the ensemble estimate is large, where darker shows higher confidence.**

The results from Figure 2(a-c) are summarised with the standard deviation of the ensemble members
over time (Figure 2d) and space (Figure 3). As noted, the AZ (Figure 2c) has the largest disagreement

amongst methods shown by map (Figure 3a) and the differences between the solid and dashed lines in
Figure 2d, particularly during summer and autumn. This is likely due to the inability of the methods to
accurately capture the larger intra-seasonal variability and patchiness in the AZ where the rapid
reduction of $p\text{CO}_2$, due to melting sea ice leads to patchy $p\text{CO}_2$ distributions (Bakker et al. 2008;
Chierici et al. 2012). The ensemble members are more coherent in the SAZ and PFZ.


In order to ascertain a degree of coherence and confidence in the ensemble we show the signal to noise
ratio in Figure 3c. We define the signal as the largest difference in the trend for a particular point. This
is calculated from the largest difference of annual averages of $\Delta p\text{CO}_2$. The noise is the mean standard
deviation of ensemble member estimates. A large signal to noise ratio (Figure 4c) is indicative of a large

trend signal compared to the variability of the ensemble. While signal to noise ratio is > 1 for the entire





domain, there are regions where the ratio is < 2: parts of the Atlantic sector of the SAZ and the Indian sector of the PFZ.

With the established baseline of confidence in the ensemble, the ensemble mean of $\Delta p\mathrm{CO}_2$ can be assessed. The seasonal cycle is the strongest mode of $\Delta p\mathrm{CO}_2$ variability in the Southern Ocean (Lenton et al. 2013). It is therefore important that the ensemble mean is understood in the context of our current understanding of the seasonal cycle.

## 3.2 Ensemble seasonal cycle

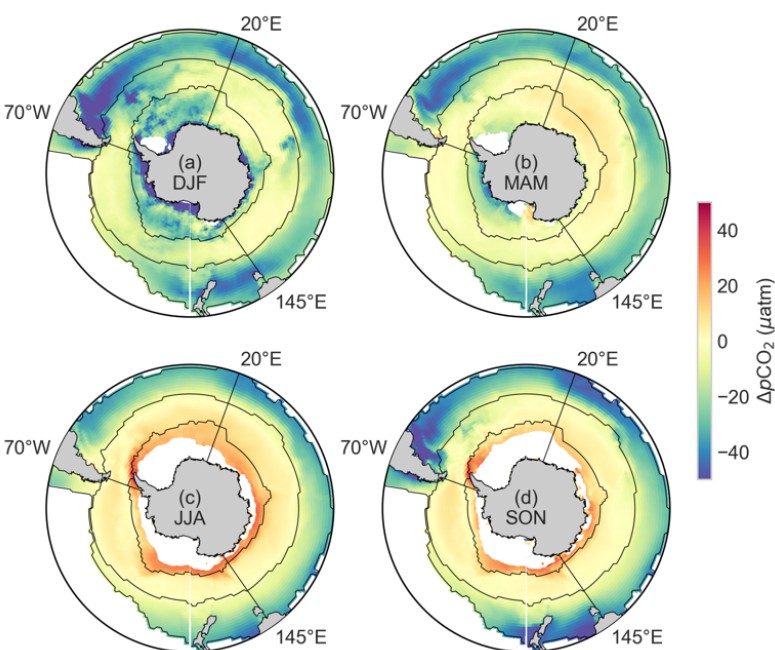

Figure 4: The mean seasonal states of $\Delta p\mathrm{CO}_2$ of the empirical ensemble mean]{Seasonal averages of $p\mathrm{CO}_2$ for the ensemble of $\Delta p\mathrm{CO}_2$. These are shown for (a) summer, (b) autumn, (c) winter and (d) spring. The black contour lines show the SAZ, PFZ and AZ from north to south as defined by Fay and McKinley (2014).

The seasonal cycle of the $\Delta p\mathrm{CO}_2$ for each biome (Figure 2a-c and Figure 4a-c) is coherent with expected seasonal processes based on literature (Metzl et al. 2006; Thomalla et al. 2011; Lenton et al.

2012; Lenton et al. 2013). In all biomes, uptake of $\mathrm{CO}_2$ is stronger during summer than in winter giving rise to the strong seasonal cycle. This is due to the opposing influences of the dominant winter and summer drivers, partially damped by the seasonal cycle of temperature (Takahashi et al. 2002; Thomalla et al. 2011; Lenton et al. 2013). The dominant processes of mixing and entrainment in winter results in increased surface $p\mathrm{CO}_2$ and thus outgassing (Takahashi et al. 2009; Lenton et al. 2013;

Rodgers et al. 2014). In summer, stratification also allows for increased biological production and the consequent uptake of CO2, thus reducing the entrained winter DIC and associated $p\mathrm{CO}_2$ (Bakker et al. 2008; Thomalla et al. 2011). However, stratification typically limits entrainment, but does not exclude





the occurrence of entrainment during periods of intense mixing driven by storms, which has an impact
on both primary productivity, DIC and $p$CO$_2$ (Lévy et al. 2012; Monteiro et al. 2015; Nicholson et al.
2016; Whitt et al. 2017).

The SAZ (Figure 2a) is a continuous sink where summer uptake (Figure 4a) is enhanced by biological
production and winter (Figure 4c) mixing results in a weaker sink (Metzl et al. 2006; Lenton et al. 2012;
Lenton et al. 2013). These processes produce a similar seasonal amplitude in the PFZ (Figure 2b), but
the seasonal fluxes are opposing: a sink in summer (< 0 µatm) and a source in winter (> 0 µatm).
However, this is according to the mean state in the PFZ and winter estimates of $\Delta p$CO$_2$ do in fact
approach 0 µatm toward the end of the time series (Figure 2b). The AZ has the strongest seasonal cycle
due to upwelling of CO$_2$ during winter and strong biological uptake in summer. However, much of this
is dampened by sea ice cover during winter and weaker winds during summer (Ishii et al. 1998; Bakker
et al. 2008).

It is important to note that $\Delta p$CO$_2$ is zonally asymmetric within each biome, particularly during summer
(Figure 4a) when biological uptake of CO$_2$ increases. Zonal integration of $\Delta p$CO$_2$ and $F$CO$_2$ could thus
dampen regional signals. A regional approach is therefore needed to examine the regional
characteristics of seasonal and interannual variability of $\Delta p$CO$_2$ and $F$CO$_2$ and to understand their
drivers.

### 3.3 Regional $p$CO$_2$ and $F$CO$_2$ Variability: Zonal and basin contrasts

Here $\Delta p$CO$_2$ and $F$CO$_2$ are decomposed into nine domains by biome and basin with the boundaries
shown in Figure 1. The data are plotted as time series for $p$CO$_2$ (Figure 5) and $F$CO$_2$ (Figure 6)
showing: the mean annual trends of $p$CO$_2$ and $F$CO$_2$ (black lines), the maximum winter values (red
line) and the projected summer minima (dashed red line) based on adjusting the winter maxima each
year by the mean of the difference between the winter maxima and the summer minima (Figures 5, 6).
The projected summer minima implies that there is an expectation that summer $\Delta p$CO$_2$ is dependent on,
but not restricted to, the baseline set by the winter maxima. Differences between the ensemble summer
minima and projected minima are highlighted with green and blue patches, highlighting periods of
decoupling between summer and winter interannual variability. The green areas indicate periods of
strong uptake (relative to winter) that enhance the mean uptake of CO$_2$. Conversely, blue areas show
periods where weak summer uptake (relative to winter) offsets winter trend, thus reducing the mean
trend.



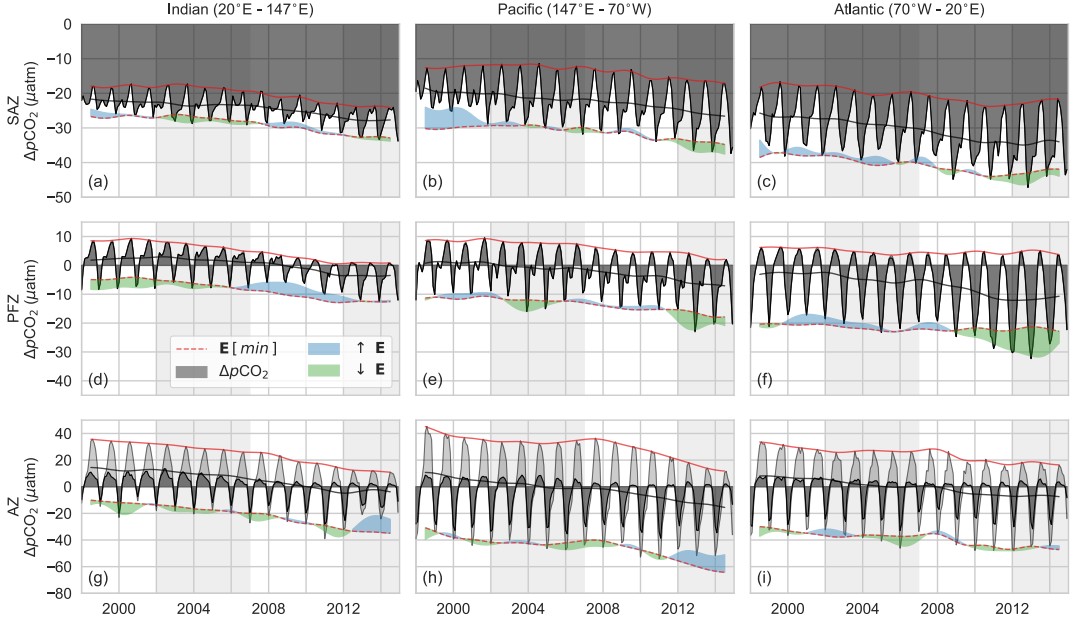

**Figure 5: Figures (a-i) show ΔpCO₂ (dark grey) and (j-r) show FCO₂ (dark grey) plotted by biome (rows) and basin (columns). Biomes are defined by Fay and McKinley (2014). The solid red line shows the maximum for each year (winter outgassing) and the dashed line shows the same line less the average difference between the minimum and maximum – this is the expected amplitude. The shaded blue (green) area shows when the annual minimum is less (greater) than the expected amplitude. ΔpCO₂ for plots (g–i) was normalised to sea ice cover, but under ice ΔpCO₂ estimates were still used to find the expected amplitude. Light grey shading in (a-i) shows the proposed periods used in Figure 9 andFigure 10 Light grey shading in (j-r) shows the "saturation" period (1998 to 2001) and the "reinvigoration" period (2002 to 2011).**

The data for $\Delta p\text{CO}_2$ and $F\text{CO}_2$ (Figures 5 and 6), show that the Southern Ocean sink strengthened from 2002 to 2011 in all domains, a period identified as the *reinvigoration* by Landschützer et al. (2015). This was preceded by a period of a net weakening sink in the 1990's referred to as the *saturation* period after Le Quéré et al. (2007). These two periods are highlighted by the grey fill in Figure 6. The saturation of the Southern Ocean CO₂ sink is not as strong in the ensemble, occurring in only five of the nine domains (see the positive trend slopes between 1998 and 2002 in Figure 6). In the last period (2012 to 2014) of the ensemble three domains go from growing uptake to reducing uptake; however, our confidence in the increasing trends from 2012 to 2014 is low due to only three years of data, with very sparse data in 2014.

Importantly, these interannual trends are the integrated effect of decoupled seasonal modes of variability. This is particularly evident in the PFZ (Figures 5d-f). Here, and in the other biomes, there is a strengthening of the sink due to a reduction of $\Delta p\text{CO}_2$ in winter on roughly a decadal mode. This corresponds with the findings of Landschützer et al. (2016), who linked the decadal variability to the Southern Annular Mode (SAM) – the dominant mode of atmospheric variability in the Southern Hemisphere (Marshall, 2003). In comparison, summer $\Delta p\text{CO}_2$ variability is sub-decadal or roughly 4 – 6 years, thus impacting the short term variability of the annually integrated trend. This is demonstrated well in the Indian sector of the PFZ where a decrease in winter $\Delta p\text{CO}_2$ from 2002 to 2011 is offset by



weakening of the summer sink from 2006 to 2010 (Figure 5d, 6d). Similarly in the Atlantic and Pacific sectors of the SAZ and PFZ strong decoupling occurs from ~2011 to the end of 2014 with a rapid increase in the strength of the summer sink.

The mean amplitude of the seasonal cycle of $\Delta p CO_2$, the mean difference between the summer minima

and the winter maxima, is perhaps a better means of understanding the magnitude of the seasonal drivers for each domain than the mean $\Delta p CO_2$ (Table A1). For example the Atlantic sectors of the SAZ and PFZ (Figures 5c,f) have the strongest seasonal variability (20.57 and 27.67 µatm respectively). This contrasts the relatively weak seasonal amplitude in the Indian sector of the Southern Ocean which has mean amplitudes of 8.85 and 13.31 µatm for the SAZ and PFZ respectively (Figures 5b,e). This contrast

can also be seen by comparing the mean seasonal maps of $\Delta p CO_2$ in Figures 4a and 4c. In summer, strong uptake in the eastern Atlantic sector of the southern ocean is indicative of large biological drawdown of $CO_2$ by phytoplankton (Thomalla et al. 2011). Conversely, relatively low primary production in the Indian sectors of the SAZ and PFZ result in a small seasonal amplitude (Thomalla et al. 2011). This large discrepancy in biological primary production is related to the availability of iron, a

micronutrient required for photosynthesis. The lack of large land masses, a source of iron, in the Indian sector of the Southern Ocean could be a contributing factor to the lack of biomass (Boyd and Ellwood, 2010; Thomalla et al. 2011).

The seasonal amplitude in the AZ is much larger due to strong contrast of the upwelling of $CO_2$ rich

deep water contained beneath winter sea ice and the strong biological drawdown in the beginning of summer (Ishii et al. 1998, Bakker et al. 2008). Rapid stratification and iron supply by melting sea ice provide the environment for phytoplankton to proliferate in the AZ. This results in large seasonal amplitudes of 46.72, 75.46 and 64.29 µatm for the Indian, Pacific and Atlantic.

The air – sea fluxes of $CO_2$ ($FCO_2$) have decadal trends that are coherent with the $pCO_2$ (Figure 6), but there are notable differences that emerge from the impact of wind on the rate of exchange as well as the surface area of each domain (Figure A1). Most prominent are the changes in the seasonal cycle and the mean seasonal sink of $FCO_2$ relative to $pCO_2$ with amplification in the Indian sector (Figures 5a,d and 6a,d) and weakening in the Atlantic Ocean (Figures 5c,f and 6c,f) of the Southern Ocean. The Indian

sector of the SAZ (Figure 6a) dominates the uptake of $CO_2$ with an annual mean flux of -0.25PgC yr-1 compared to -0.19 and -0.17 PgC yr-1 for the Atlantic and Pacific sectors respectively (where the latter are significantly different with p = 0.01). The seasonality of wind stress (see Figure A1) results in a damped seasonal cycle of $FCO_2$ in the SAZ and increasing intra-seasonal variability (compared to $\Delta p CO_2$), with stronger winter winds compensating for a weaker $\Delta p CO_2$ gradient (Monteiro et al. 2015).




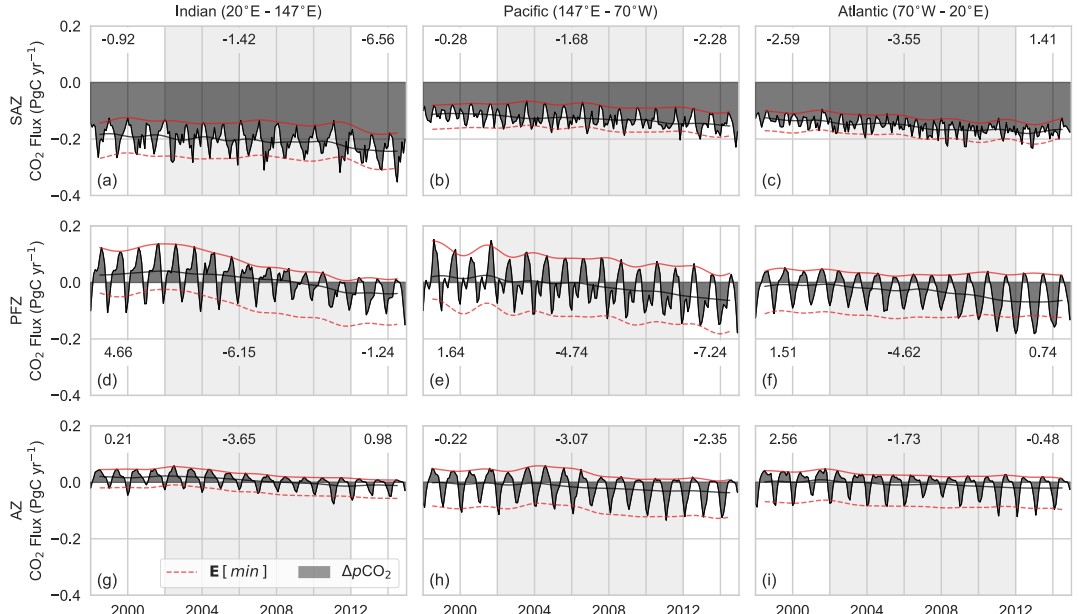

**Figure 6: Figures (a-i) show $\Delta p$CO$_2$ (dark grey) and (j-r) show $F$CO$_2$ (dark grey) plotted by biome (rows) and basin (columns). Biomes are defined by Fay and McKinley (2014). The solid red line shows the maximum for each year (winter outgassing) and the dashed line shows the same line less the average difference between the minimum and maximum – this is the expected amplitude. The shaded blue (green) area shows when the annual minimum is less (greater) than the expected amplitude. $\Delta p$CO$_2$ for plots (g–i) was normalised to sea ice cover, but under ice $\Delta p$CO$_2$ estimates were still used to find the expected amplitude. Light grey shading in (a-i) shows the proposed periods used in Figure 9 and Figure 10 Light grey shading in (j-r) shows the "saturation" period (1998 to 2001) and the "reinvigoration" period (2002 to 2011).**

This contrasts the PFZ, where opposition of summer uptake and winter outgassing of CO$_2$ is amplified by stronger wind stress (Figure 6d-f). Interannual variability is also enhanced, particularly during winter in the Indian sector of the PFZ, where a reduction in outgassing of 0.18 PgC yr$^{-1}$ is observed. The decoupling between summer and winter $F$CO$_2$ also becomes more pronounced in this region (Figure 6d), resulting in a lag in the decreasing trend. In other words, the trend of $F$CO$_2$ for the reinvigoration (2002 through 2011: -10.85 PgC yr-1) would have been stronger if the decoupling had not occurred. Similarly, the seasonal decoupling in the Pacific sector of the SAZ and PFZ results in a stronger growing sink from 2012 to 2014. In the Atlantic sector of the SAZ and PFZ the earlier onset of the seasonal decoupling (Figure 5c,f) also means that re-coupling occurs sooner, resulting in a positive flux trend (Figure 6c,f). Lastly, $F$CO$_2$ in the MIZ is damped during winter due to ice cover and weaker winds during summer when $\Delta p$CO$_2$ is low due to the short-lived intense biological uptake of CO$_2$ (Ishii et al. 1998; Bakker et al. 2008).

### 3.4 Seasonal deconstruction of interannual variability

Figures 5 and 6 give us insight into the magnitude of $F$CO$_2$ and $\Delta p$CO$_2$ interannual variability as well as the character of these changes; *i.e.* decoupling of decadal winter and sub-decadal summer interannual modes of variability. This alludes to the fact that $\Delta p$CO$_2$ and $F$CO$_2$ are responding to different adjustments of seasonal large scale atmospheric forcing and/or responses of internal ocean dynamics in



the Southern Ocean (Landschützer et al. 2015, 2016; DeVries et al. 2017). In the study by Landschützer et al. (2015) it was advanced that the explanation for the reinvigoration of $\Delta p\mathrm{CO_2}$ uptake in the 2000s decade was linked to the net thermal control driven by a response of DIC and temperature to asymmetric atmospheric forcing over the Southern Ocean. More recently, a study by DeVries et al.

(2017) proposed that the Southern Ocean uptake was due to the global deceleration of the Meridional Overturning Circulation (MOC). They suggested that the MOC was increasing the oceanic storage and reducing the losses of $\mathrm{CO_2}$ to the atmosphere, particularly in the Southern Ocean. The mechanism proposed by DeVries et al. (2017) is the same as that put forward by Le Quéré et al. (2007) and Lovenduski et al. (2008) amongst others, where the changes in outgassing are related to the strength of

the westerly winds over the Southern Ocean. These studies have in linked the wind stress variability to SAM.

In order to capture the decoupled short term variability observed during summer, the data are broken into four interannual periods (P1 to P4) shown by the light grey fills in Figure 5. The first period is the

saturation period (P1: 1998 – 2002) by Le Quéré et al. (2007). The second and third periods are informed by the reinvigoration period (2002 through 2011) split around the start of 2007 an early, weaker reinvigoration (P2: 2002 – 2006) and a late, stronger reinvigoration (P3: 2007 – 2011). The last period incorporates the three years of new data (P4: 2012 to 2014).

These four periods are too short for trend analyses (Fay and McKinley, 2014), but the intention here is to identify periods that are short enough to resolve interannual changes in the large scale drivers of the winter and summer mean values for $p\mathrm{CO_2}$ and $F\mathrm{CO_2}$ that would otherwise be averaged out over longer periods. We perform an anomaly analysis between each period rather than a trend analysis (for which inflections of $\Delta p\mathrm{CO_2}$ would be more suitable delimiters). The relative anomalies in the drivers are their

differences between two adjacent periods (*e.g.* P2 – P1). As a result four periods give rise to three transition anomalies for summer and winter: *A* (P2 – P1), *B* (P3 – P2) and *C* (P4 – P3). Note that although only summer and winter are discussed, it is recognised that autumn and spring could be equally mechanistically important. Winter anomalies of $\Delta p\mathrm{CO_2}$, wind stress, SST and MLD are shown in Figure 7. Summer anomalies of $\Delta p\mathrm{CO_2}$, wind stress, SST and Chl-*a* are shown in Figure 8 where

MLD, in winter, is replaced with Chl-*a* for summer as it is potentially a more important driver than the generally shallow MLD in summer (the omitted plots are shown in Figures A2 and A3).





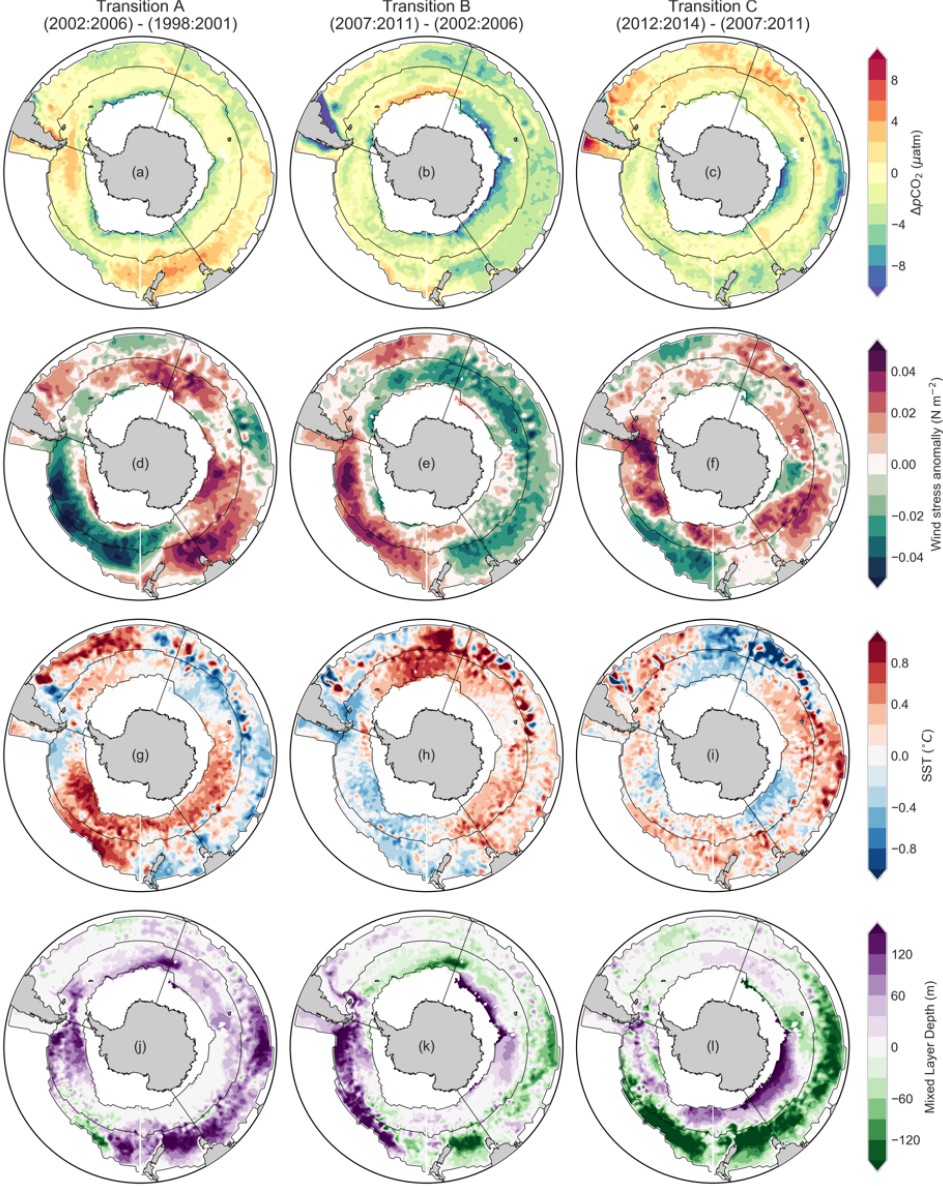

**Figure 7: Transitions of winter $\Delta p$CO$_2$, wind stress, SST and MLD]{Transitions (relative anomalies) of winter $\Delta p$CO$_2$ (a-c), wind stress (d-e), sea surface temperature (f-h) and mixed layer depth (i-k) for four periods (as shown above each column). The thin black lines show the boundaries for each of the nine regions described by the biomes (Fay and McKinley 2014) and basin boundaries.**

## 3.5 Drivers of the decadal winter trends in $\Delta p$CO$_2$ and $F$CO$_2$

There are two features of interest in the anomalies of winter $\Delta p$CO$_2$ and its drivers (Figure 7). First, there is a zonally asymmetric dipole for wind stress between the Pacific and Indian sectors of the Southern Ocean that dominates transitions A and B. Second, $\Delta p$CO$_2$, SST and MLD cohere roughly to the spatial variability of wind stress anomalies, mirroring the wind dipole. These features will be expounded on in the paragraphs that follow.



Transition A (the transition from P1, the saturation period, to P2, the start of the reinvigoration) shows a

relative increase of $\Delta p\mathrm{CO_2}$ in the east Indian and Pacific sectors of the SAZ – suggesting a delay in the onset of the reinvigoration for these basins. This regional sustained saturation corresponds to a shift towards stronger winds and deeper MLDs west of the Tasman Sea (Pacific sector of the SAZ) and surrounds (Figure 7d,j). In contrast, the Atlantic and west Indian sectors of the SAZ, and the south-eastern Indian sector of the PFZ show a reinvigoration of $CO_2$ uptake. In the central Pacific, weaker

wind stress (green in Figure 7d) corresponds with relative warming and shoaling or stagnation of MLD. This is consistent with an invigoration of the Southern Ocean $CO_2$ sink (P1 – P2: 1998 – 2007) initiated by a weakening of the mean westerly wind stress in the Pacific and W Indian Oceans, which we are suggesting, reduced winter entrainment and possibly upwelling of $CO_2$-rich deep waters (Marshall, 2003; DeVries et al. 2017).


Transition B, which corresponds to anomalies between P2 to P3 (the two reinvigoration periods), is characterized by an invigoration of $\Delta p\mathrm{CO_2}$ (-ve shift) in all basins, but particularly in the Indian basin (Figure 7b). This strengthening of the $CO_2$ uptake corresponds with weaker wind stress, a warming trend in surface waters and shoaling MLDs in the SE Atlantic and Indian Ocean sectors in the winter

(Figure 7b,e,h,k). In the Pacific, the opposing effects of the dipole are: stronger wind stress, deeper MLD, and cooler surface waters. These changes are associated with the persistence of a neutral to weak reduction of $\Delta p\mathrm{CO_2}$ compared to the Indian sector. All the changes in transitions A and B are coherent to changes in the Pacific–Indian wind stress dipole.

In transition C (P4 – P3: 2012 - 2015), when we propose the invigoration trend starts to weaken, $\Delta p\mathrm{CO_2}$ strengthens further in the northern and southern extremes of the Indian and Pacific basins, albeit in a more spatially heterogeneous way. In the Atlantic sector the previous invigoration trend reverses completely and the $CO_2$ sink in the winter is shown to weaken (Figure 7b,c). The previously well characterized Pacific–Indian dipole is not apparent, suggesting that transition A and B capture well

established phases in the decadal variability, while transition C may be capturing a transition into the following phase which could take the system back to the same configuration as P1. The notion that this period is a snapshot between two distinct phases is supported by the relatively heterogeneous spatial structure of wind stress in both the Atlantic and Indian Oceans.

### 3.5.1 Wind dominated interannual variability of $p\mathrm{CO_2}$ in winter

We propose that the interannual variability of seasonal wind stress in winter may be the dominant driver of the saturation and reinvigoration periods. Moreover, the suggested Pacific–Indian wind dipole may



be linked to the decadal variability of $\Delta p\text{CO}_2$ observed in the Southern Ocean (Landschützer et al. 2016).

Increasing or decreasing interannual winter wind stress variability impacts $\Delta p\text{CO}_2$ (and thus $F\text{CO}_2$) by driving stronger turbulent mixing. In the transition to and during winter, this mixing is associated heat loss resulting in a loss of buoyancy or weaker stratification (Abernathey et al. 2011). Weaker buoyancy facilitates deepening of the MLD, thus entraining DIC-rich deep waters (Abernathey et al. 2011; Lenton et al. 2013). Conversely, decreased wind stress and mixing during winter (on seasonal or interannual

time scales) reduces the rate of heat loss (represented as warm anomalies in Figure 7). This results in stronger stratification and shallower winter MLD limits entrainment of DIC, which strengthens the $\text{CO}_2$ winter disequilibrium and leads to a stronger $\text{CO}_2$ sink anomaly (Figure 7). This is the mechanism that results in decreasing or increasing fluxes with changes in wind. However, the direct link between this wind stress mechanism and the reinvigoration was not made by Landschützer et al. (2015). This may

be, in part, due to the seasonal decoupling that may lead to biased interpretation of wind stress and SST.

Past studies have used the SAM as a proxy for wind stress variability over the Southern Ocean, where the multi-decadal increasing trend has been cited as a reason for the saturation in the 1990's (Marshall, 2003; Le Quéré et al. 2007; Lenton and Matear, 2007; Lovenduski et al. 2008). While Landschützer et

al. (2016) identified the SAM as being a driver of global CO2 variability, the index does not explain the reinvigoration of the Southern Ocean. The SAM is often represented as a zonally integrating index (Marshall, 2003), but studies have shown that the SAM, as the first empirical mode of atmospheric variability, is zonally asymmetric (Fogt et al. 2012). The zonal asymmetry of the SAM is linked with ENSE and is strongest in winter, particularly over the Pacific sector of the Southern Ocean during a

positive phase, thus in accord with the Pacific–Indian winter wind stress dipole (Barnes and Hartmann 2010; Fogt et al. 2012). Fogt et al. (2012) noted that the SAM has become more zonally symmetric in summer since the 1980's, matching the wind stress anomalies seen in Figure 7.

DeVries et al. (2017) proposed that slowing down of Meridional Overturning Circulation (MOC) as an

alternate mechanism for the reinvigoration of the Southern Ocean $\text{CO}_2$ sink in the 2000's. The authors explain that weaker overturning reduces the natural $\text{CO}_2$ brought from the deep to the surface ocean. Moreover, they suggest that this mechanism may continue to drive intensification of the global $\text{CO}_2$ sink. The longer modes of MOC variability makes it difficult to attribute the change in flux to changes in overturning.


This poses an interesting question for the Southern Ocean carbon sink when we consider that weakening MOC may counteract the intensification of winds over the Southern Ocean (encapsulated by





the increasing SAM). Meredith et al. (2012) found that this question is made more complex by the compensatory effect of increased eddy activity (measured by eddy kinetic energy – EKE) to enhanced

northward Ekman transport driven by intensified winds (Meredith and Hogg, 2006; Abernathey et al. 2011; Marshall and Speer, 2012). Moreover, the inclusion of these eddies in a high resolution model reduced $CO_2$ outgassing driven by increased Ekman transport by one third by entraining alkalinity to the surface water (Dufour et al. 2013). As it stands, this is an unresolved question and more work will have to be undertaken to understand the effect of these two counteracting mechanisms of $CO_2$ transport.


In summary, we propose that interannual variability of wind stress is the dominant interannual driver of $F\text{CO}_2$ in the Southern Ocean. The interannual variability of wind stress is linked to the SAM, but this relationship is nuanced by the zonally asymmetric variability of the SAM.

**3.6 Trends in the anomalies of $\Delta p\text{CO}_2$ and its drivers in summer**

The most important difference between the summer and winter anomalies, is that wind stress (Figures 8d-f) does not correlate to $\Delta p\text{CO}_2$ (Figures 8a-c), thus ruling out wind as a first order driver of summer $CO_2$. Rather, $\Delta p\text{CO}_2$ has the strongest coherence with Chl-*a* (an inverse relationship), which suggests that primary production may be a first order driver of the observed $\Delta p\text{CO}_2$ variability.

Looking more specifically at the variability of $\Delta p\text{CO}_2$, Transition A (P2 – P1 in Figure 8a) is marked by

patchy decreases $CO_2$ in regions of high EKE (Agulhas retroflection and Tasman shelf) in the SAZ and AZ, mirrored by an increase in Chl-a. The Atlantic and Indian sectors of the PFZ remain neutral/weak sources marked by a reduction in phytoplankton biomass (Figure 8j). Transition B (P3 – P2 in Figure 8b), shows invigoration of $CO_2$ uptake in: the Atlantic sector of the SAZ and PFZ; the Indian sector of the AZ; and patchy strengthening in the Pacific Ocean. Once again, the reduction of $\Delta p\text{CO}_2$ from P2 to

P3 in the aforementioned regions correlate well with Chl-a. In transition C (P4 – P3 in Figure 8c) the reduction of the $\Delta p\text{CO}_2$ is widespread in the Indian and Pacific Oceans in all three biomes, though this does not necessarily correspond with Chl-a. There is a strong decrease in Chl-a and concomitant increase in $\Delta p\text{CO}_2$ along Polar front in the Atlantic sector, coinciding with position of the ACC which has high EKE (Meredith, 2016). These examples demonstrate that $\Delta p\text{CO}_2$ is driven primarily by Chl-a

in summer. However, understanding Chl-a variability is perhaps more complex as there is seemingly no set rule between Chl-a, SST and wind stress (Thomalla et al. 2011).



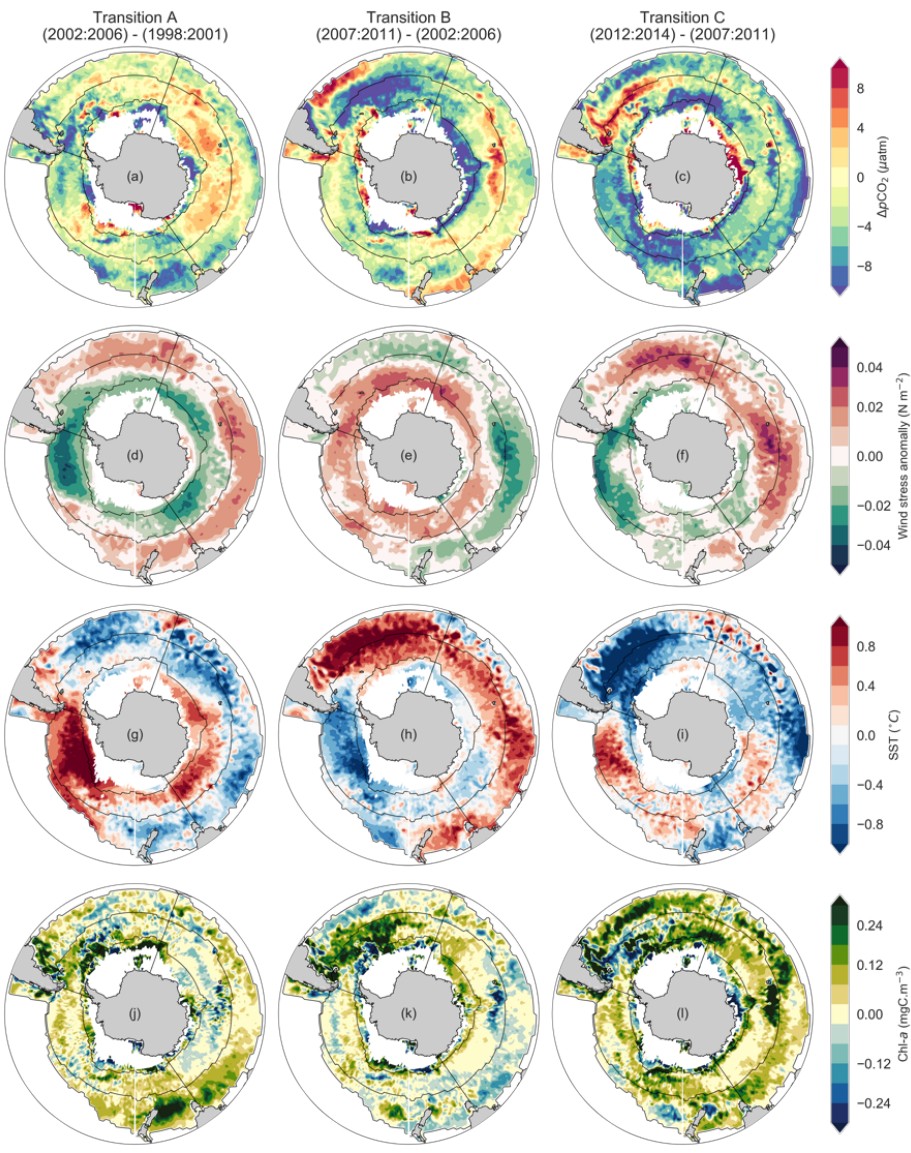

**Figure 8: Transitions of summer ΔpCO₂, wind stress, SST and Chl-*a*]{Relative anomalies of summer ΔpCO₂ (a-c), wind stress (d-e), sea surface temperature (f-h) and mixed layer depth (i-k) for four periods (as shown above each column). The thin black lines show the boundaries for each of the nine regions described by the biomes (Fay and McKinley 2014) and basin boundaries.**


There are regions in the Southern Ocean where summer Chl-a variability does not coincide with $\Delta p$CO$_2$ variability, particularly in the Indian and Pacific sectors of the SAZ (Figures 8a-c and 8j-l). This may be due to the fact that chlorophyll concentrations, and anomalies thereof, are low in these regions (Thomalla et al. 2011). This may result in the other variables, SST and wind stress, becoming higher

order drivers as found by Landschützer et al. (2015) and Munro et al. (2015).

It is thus important to understand the variability of SST and wind stress in summer. Large SST anomalies between the western Atlantic and eastern Pacific sectors vary as an zonally asymmetric



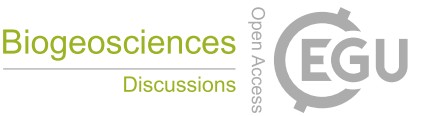

dipole. As in winter, there is a summer wind stress anomaly dipole, but rather than being zonally
asymmetric (*e.g.* Pacific–Indian), the dipole has annular, north-south variability (Figures 7,8d-f). We
suggest that these dipoles in the variability may indicate that the Southern Ocean, as a system,
transitions between different states forced by atmospheric variability (Landschützer et al. 2015).

Lastly, there is seasonal variability in the magnitudes of ΔpCO2 and the drivers. The anomalies of
ΔpCO2 and SST have a larger magnitude than the winter anomalies. Conversely, the wind stress
anomalies are for winter than in summer. This is an important consideration for analyses that aim to
understand the driving mechanisms, where annual averaging would make it difficult to decompose the
true drivers of change.

### 3.6.1 Chlorophyll dominated interannual anomalies of $p$CO$_2$ in summer

The fact that Chl-a is the dominant driver of interannual $\Delta p$CO$_2$ variability should not be surprising
given that models and observations support this (Hoppema et al. 1999; Bakker et al. 2008; Mahadevan
et al. 2011; Wang et al. 2012; Hauck et al. 2013; 2015; Shetye et al. 2015). However, our data show that
the dominance of interannual Chl-a variability over $\Delta p$CO$_2$ is largely limited to regions where Chl-a is
high, such as the Atlantic, the Agulhas retroflection and south of Australia and New Zealand (Figure 9).

However, the spatial variability of high Chl-a regions in the Southern Ocean is complex due to the
dynamics of light and iron limitation (Arrigo et al. 2008; Boyd and Ellwood 2010; Thomalla et al. 2011;
Tagliabue et al. 2014; 2017). This complexity is highlighted in Thomalla et al. (2011), where the Chl-a
is characterized into regions of concentration and seasonal cycle reproducibility (Figure 9). The
seasonal cycle reproducibility (SCR) is calculated as the correlation between the mean annual seasonal
cycle and the observed chlorophyll time series. Here we use the approach of Thomalla et al. (2011), in
Figure 9, as a conceptual framework to understand the interannual variability of $\Delta p$CO$_2$.

### 3.6.2 High chlorophyll regions

While regions of high SCR (dark green in Figure 9) do not correspond with the interannual variability
of Chl-*a* (Figure 8j-l), the framework by Thomalla et al. (2011) does present a hypothesis by which the
variability of Chl-*a* and its drivers can be interpreted. This is, that the variability of Chl-*a* in a region is
a complex interaction of the response of the underlying physics (mixing vs buoyancy forcing, which
modulate light (MLD) and iron supply, to the interannual variability in the drivers (SST and wind
stress). This complexity is exemplified by strong warming in the Atlantic during transition B, which
results in both an increase and decrease in Chl-*a*, with inverse consequences for $\Delta p$CO$_2$. The effect is
even stronger transition C, where strong cooling in the Atlantic results in both a decrease and increase
of Chl-*a* (Figure 8i,l). In both transition A and B the respective increase and decrease of Chl-*a* occur





roughly over the ACC, while the opposing effects during transitions A and B occur roughly to the north

and south of the ACC region. These temperature changes may impact the stratification of the region, but

complex interaction with the underlying physics results in variable changes in Chl-*a*.

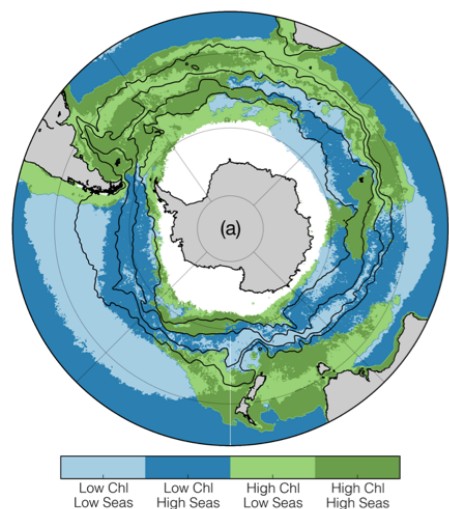

| Low Chl | Low Chl | High Chl | High Chl |
| Low Seas | High Seas | Low Seas | High Seas |

**Figure 9: Chl-*a* seasonal cycle reproducibility and iron supply mechanisms in the Southern Ocean]{(a) Regions of chlorophyll biomass and seasonal cycle reproducibility from Thomalla et al. (2011) (using SeaWIFS data). Seasonality is calculated as the**
**correlation between the mean annual seasonal cycle compared to the observed chlorophyll time series. A correlation threshold of 0.4 was applied to each time series to distinguish between regions of high and low seasonality; similarly, a threshold of 0.25 mg m$^{-3}$ was used to distinguish between low or high chlorophyll waters. Black lines showing the fronts are the same as described in figure.**

It is clear that while there is a relationship between Chl-a and $p$CO$_2$ as well as a relationship between

wind stress and SST in summer, the relationship between wind forcing and Chl-a and $p$CO$_2$ is not as

strong as in the winter anomalies (Figure 7). The reason for this is likely to be that enhanced summer

buoyancy forcing resulting from summer warming and mixed layer eddies drives a more complex

response to wind stress in the form of vertical velocities (w) and mixing (Kz), which influence the iron

supply and the depth of mixing (McGillicuddy, 2016; Mahadevan et al. 2012).

Mesoscale and sub-mesoscale processes may have a part to play in these dynamic responses of Chl-*a* to

changes in SST and wind stress (amongst other drivers). For example, eddy-driven slumping is a sub-

mesoscale process that acts to rapidly shoal the mixed layer (Mahadevan et al. 2012). This allows

phytoplankton to remain within the euphotic zone and thus grow (while iron is not limiting). Similarly,

Whitt et al. (2017) demonstrated that a combination of high and low frequency oscillation of down-front

winds are able to enhance nutrient entrainment (including iron) into the mixed layer on the less dense

side of a front. This has important implications for Southern Ocean fronts, where Chl-*a* may benefit

from this entrainment mechanism combined with eddy-driven slumping that could subsequently rapidly

shoal the mixed layer (Du Plessis et al. 2017).



Storm driven, intra-seasonal mixing is another sub-mesoscale process that could alleviate iron limitation through shear-driven mixing along the base of the mixed layer (Nicholson et al. 2016). Importantly, both storm driven entrainment and the oscillatory enhancement of entrainment, rely on a mixing transition layer that has sufficient iron that is able to sustain growth – weak dissolved iron gradients in the Pacific and east Indian sectors of the Southern Ocean would explain the lack of phytoplankton in

these regions (Tagliabue et al. 2014; Nicholson et al. 2016). Much of the spatial character of the transition anomalies occurs at mesoscale, which strengthens the view that these mesoscale and sub-mesoscale processes may be key to explain changes in Chl-*a* (Figure 8j-l). This level of mechanistic detail was not part of this study.

### 3.6.3 Low chlorophyll regions

Entrainment and stratification can explain much of the variability in the eastern Pacific and Indian sector of the PFZ (with the exception of the wake of the Kerguelen Plateau). For example, in the eastern Pacific in transition A (Figure 8a,d,g), strong warming and weaker winds have little impact on Chl-*a*, but a decrease in $\Delta p\mathrm{CO_2}$ is observed. Conversely, cooling in the west Indian sector of the PFZ results in a weak increase in $\Delta p\mathrm{CO_2}$ during the same transition. In both these cases, the effect of cooling or

warming on $\Delta p\mathrm{CO_2}$ is negligible relative to the impact of entrainment or stratification respectively. The effect is reversed in the eastern Pacific during transition B where strong cooling results in a weak reduction of $\Delta p\mathrm{CO_2}$ rather than the increase that would be expected from entrainment. This is the mechanism that Landschützer et al. (2015) describe in the Pacific, where enhanced entrainment of DIC and TA is compensated for by cooling. This emphasises that the balance between SST (as a driver of

stratification) and wind stress is far more important than in winter.

In summary, regions with high biomass, Chl-*a* integrates the complex interactions between SST, wind stress, MLD and sub-mesoscale variability resulting in large interannual $p\mathrm{CO_2}$ variability compared to low biomass regions. In low Chl-*a* regions, wind driven entrainment/stratification are in general

dominant over thermally driven changes of $\Delta p\mathrm{CO_2}$.

### 4 Synthesis

In this study, ensemble means of empirically estimated $\Delta p\mathrm{CO_2}$ and $F\mathrm{CO_2}$ were used to investigate the trends and the drivers of these trends in the Southern Ocean. The ensemble mean of $\Delta p\mathrm{CO_2}$ showed that the seasonal cycle is the dominant mode of variability imposed upon a weaker interannual and decadal

trends. The data were separated into nine domains defined by functional biomes and oceanic basins to account for the roughly basin scale zonal asymmetry observed in preliminary analyses of $\Delta p\mathrm{CO_2}$ (Fay and McKinley, 2014). A seasonal decomposition was applied to the nine domains, revealing that winter and summer interannual trends are decoupled for each region. The decadal trend was in accordance



with recent studies showing a saturation of the Southern Ocean $CO_2$ sink in the 1990's followed by the

reinvigoration in the 2000's (Le Quéré et al. 2007; Landschützer et al. 2015).

We suggest that changes in the characteristics of the seasonal cycle define the interannual and decadal modes. The mechanisms that drive the interannual and decadal modes are therefore embedded in the seasonal cycle. We propose that $\Delta pCO_2$ decadal variability is driven by primarily by changes in the

winter wind stress which influences the resulting convective entrainment of deep DIC-rich water masses (Lenton et al. 2009; 2013). This mechanism is strongest in winter due to the role of large seasonal net heat losses on convective overturning of the water column. The $\Delta pCO_2$ winter trends, agreed with wind stress variability, where the latter corresponds with the decadal variability associated with the Southern Annular Mode (Lovenduski et al. 2008; Fogt et al. 2012; Landschützer et al. 2016).


Our findings show that interannual summer variability of $\Delta pCO_2$ occurs from a baseline set by an interannual winter trend but the shorter time-scale summer linked interannual variability of $\Delta pCO_2$ (roughly 4 – 6 years) was driven primarily by Chl-a. Wind stress and sea surface temperature still influenced $\Delta pCO_2$ in summer, but were lower order drivers. We propose that the interannual variability

of the summer seasonal peak is linked to the complex interaction of mid-latitude storms with the strong mesoscale and sub-mesoscale gradients in the Southern Ocean.

We propose that there needs to be a more concerted focus through in situ and modelling experiment towards understanding the mechanisms that drive seasonal and intraseasonal variability in order to

improve the ability of ocean and earth systems models in reflecting and predicting decadal and interannual modes.





## A Additional Materials

### A1 Mean $\Delta p\text{CO}_2$ for Southern Ocean domains

The SAZ (Figures 5a-c) accounts for the majority of uptake in the Southern Ocean with a mean $\Delta p\text{CO}_2$ of -25.31 µatm. However, there is a relatively large difference between the three sectors of the SAZ, with mean $\Delta p\text{CO}_2$ values of -24.39, -22.24 and -30.48 µatm for the Indian, Pacific and Atlantic respectively. In the PFZ (Figures 5d-f) the sink is far weaker due to the opposing summer uptake and winter outgassing, with $\Delta p\text{CO}_2$ values of 0.23, -2.06 and -6.57 again in the respective order Indian,

Pacific and Atlantic. Similarly, in the AZ (Figures 5g-i) mean estimates of $\Delta p\text{CO}_2$ are muted by opposing seasonal signals with mean estimates of 4.83, -3.01 and -0.76 µatm.

Table 2: Mean $\Delta p\text{CO}2$ for each of the Southern Ocean domains shown in Figure 5, where the domains are defined according to Figure 1.

| BIOME | Indian | Pacific | Atlantic |
|---|---|---|---|
| **SAZ** | -24.39 | -22.25 | -30.48 |
| **PFZ** | 0.23 | -2.06 | -6.57 |
| **AZ** | 4.83 | -3.01 | -0.76 |


### A2 Wind speed and regional surface area

The regional magnitude of integrated air-sea $\text{CO}_2$ fluxes are in part determined by the wind speed and surface area of the specific region. Figure 11a shows the average wind speeds for summer and winter for each of the regions as defined in Figure 1. The wind product used is CCMP v2 (Atlas et al. 2011).

Figure 11b shows the surface area of each of the regions. Note that the Indian sector of the PFZ has both the highest average wind speed and has the largest surface area. This explains the dominance of the region in the determination of interannual trends of $F\text{CO}_2$, even though $\Delta p\text{CO}_2$ trends are relatively weak.

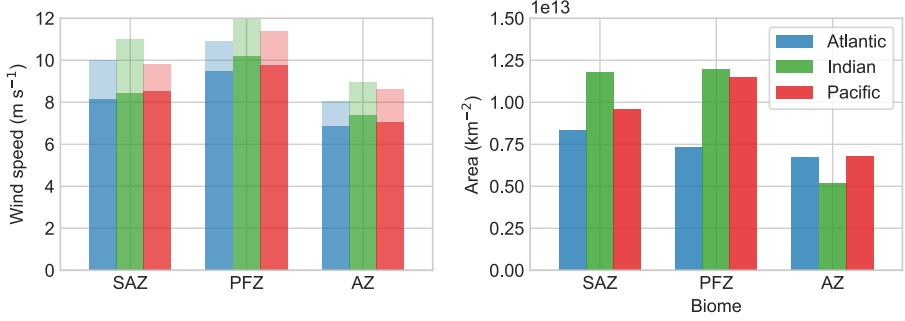

Figure A1: (a) Average wind speeds for each of the bimoes for summer (dark) and winter (light). The ocean basins are shown by the colors as shown in the key for (b). (b) shows the size of each region seperated by biome and basin.




## A3 Additional transition anomaly figures

Figure A2 and A3 augment Figures 7 and 8 respectively. These were omitted from the main text figures as we found that these variables (Chl-*a* in winter and MLD in summer) do little to aid our understanding

of changes in $\Delta p CO_2$. These are included for the sake of completeness.

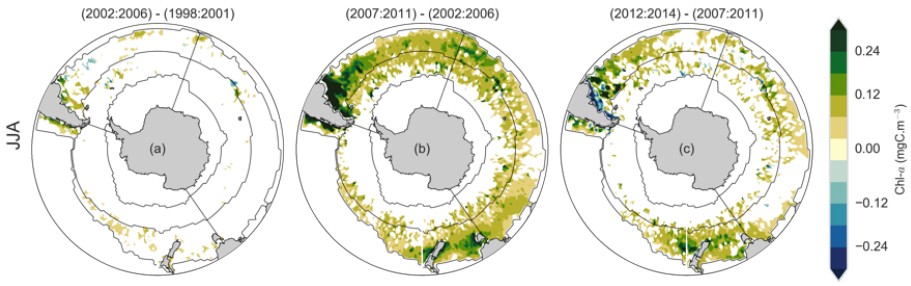

**Figure A2: Winter Chl-a transitions for each of the three anomaly periods (relating to Figure 7).**

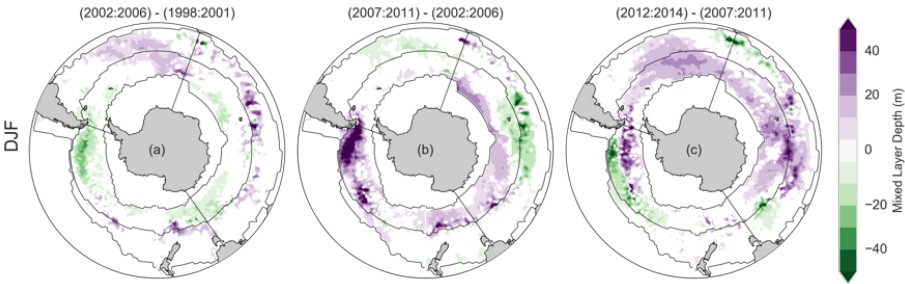

**Figure A3: Summer MLD transitions for each of the three anomaly periods (relating to Figure 8). Note that the scale of MLD in**
**this figure does not match the scale of MLD in Figure 7.**





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
