# Peer review of "Interannual drivers of the seasonal cycle of CO2 in the Southern"

_Biogeosciences, 2017_

## Referee Comment (RC1) · Anonymous Referee #1 · 5 Oct 2017

Gregor and co-authors investigate the variability of the Southern Ocean CO2 uptake strength from 1998-2014 analyzed for 9 regions (SO divided in basins and Fay and McKinley biomes). The authors combine 5 realizations to form a multi-model mean which is used to investigate seasonality and year-to-year variability of the delta pCO2 and the CO2 flux. 3 of the realizations are independent whereas 2 are simply higher resolution versions of 2 other methods. The authors find that the seasonal variability is the strongest mode of variability in the SO. Additionally, the authors confirm results from past studies that the SO was losing some of its uptake capacity in the early years of their analysis period, then the uptake increased in the subsequent period, whereas in the final years of their analysis period the reinvigoration of the sink stopped again. The authors investigate the cause of this variability in the sink strength by analyzing

anomaly periods for winter and summer season separately. From this analysis the authors conclude that the drivers are seasonally decoupled with wind being the dominant driver for winter variability and biology being the dominant driver for summer variability.

I found this study to be interesting, comprehensive and in general suitable for the journal. The authors do not only present results from new methods to confirm previous results, they also deepen the analysis by looking at anomalies rather than trends (as previously done) and investigate different seasons.

I do however have some major issues with the presented manuscript:

The study confuses trends and variability. When do the authors talk about trends and when about variability? At the moment, these two terms are mixed up. E.g. take figure 5 all panels. There is clearly some year-to-year variability causing in some years more and in some less uptake but overarching in all panels of figure 5 and 6 one can clearly see an increasing $CO_2$ sink from 1998 onwards, i.e. an increasing trend throughout the entire time period. Furthermore, wording used like "decadal trends" and "interannual trends" contribute to the confusion. What is a decadal trend? Is it the slope of a regression line when considering at least 10 years of data? What are interannual trends? The same only considering 3 years? In the latter case one cannot speak of a trend at all.

Despite being able to do so, the authors do not add uncertainties. There are many sources of uncertainty ignored by the authors, e.g. the measurement uncertainty (which is however neglectably small), the extrapolation error of the method, the building of the multi-model mean creates an error and finally the calculation of the air-sea flux adds another source of uncertainty (through wind and transfer velocity choice). At the moment, the results are presented overconfidently. It is not clear how much of the explored variability is significant and how much is simply statistical jibberish. I am aware that there is no "perfect" way to represent all uncertainties, but in a data sparse region like the Southern Ocean a study like this needs to add the best possible uncertainty

estimate, otherwise, many of the conclusions drawn cannot simply be accepted.

On page 11 line 312 I found that the authors claim statistical significance between the mean uptakes, reporting a p-value. It is not clear to the reader what test was used and how significance has been determined. Also, when adding uncertainty, the authors will notice that an uptake of -0.17 PgC/yr is unlikely to be identified as statistically significantly different from -0.19 PgC/yr in the data sparse Southern Ocean.

Despite the uncertainty of the CO2 itself there are other sources of concern related to uncertainty. Chlorophyll is e.g. also presented without discussing uncertainty. How is cloud coverage and missing chlorophyll data effecting the results? Also, wind products have been shown to have different trends locally in the Southern Ocean. This has not been mentioned.

I am also wondering to what extend the use of different products hampers the conclusions of the manuscript. SST from Reynolds is based – to the extent of my knowledge – from satellite and in-situ data, whereas ECCO MLD is from an assimilation model. I would expect some disagreement between these products that have nothing to do with "real world" disagreement. This is not a massive concern, but certainly needs to be mentioned as well.

Another source of concern is the length choice of periods P1-P4. Periods P1-P3 are all of the same length, whereas P4 is substantially shorter. The authors claim that substantial year-to-year variability occurs on various timescales (4-6 year and 10 year modes respectively), hence the variation of periods aliases this analysis.

All the above points raised are of major concern and must be addressed before the manuscript can be considered for publication.

Minor comments:

Title: the title only mentiones CO2 fluxes whereas in the manuscript largely discusses delta pCO2

Line 29: "paucity of observations (Landschutzer et al 2015)" – This is not a good reference. Bakker et al 2016 (the SOCATv2 reference) would be a better reference for such a paucity.

Line 62: remove 2nd that

Line 95: wrong reference (Bakker et al 2016)

Table 1: How has the RMSE for the 0.25x0.25 degree product been calculated? SO-CAT offers a gridded product but on 1x1 degrees. Did the authors grid the 0.25 product themselfes? If yes, has this been done the "SOCAT" way, i.e per cruise weighted or differently? Same with the 16-day timestep. More information is needed here.

Line 130: How different would the results be if a different transfer velocity was chosen? This may have significant effects on the uncertainty.

Line 134: where are [ice] data from? I am missing a reference here.

Line 175-176: "attributed this difference to the clustering step used by the SOM-FFN that created large discrepancies in the Atlantic sector." I have not found any convincing evidence for this in any of the cited papers.

Line 236: Either the authors used the wrong wording or there is a misunderstanding, but I do not see where the authors find that the delta pCO2 is zonally asymmetric within each biome. It looks like the other way around: Figure 4 looks like there is a strong zonal symmetry (besides indeed in panel a).

Lines 249-254: This paragraph is not clear. Please rephrase

Line 273: interannual trend – what is that? Interannual variability. A 3-year trend? A trend that changes sign every year?

Lines 275: decadal mode: How are you able to say this. You have 1998-2014 data , i.e. 17 years of data. How can you detect a decadal mode from such a short timeseries?

Line 305: "decadal trend" – what is this? A trend of at least 1 decade?

Lines 311-312: "-0.19 and -0.17 PgC yr-1 for the Atlantic and Pacific sectors respectively (where the latter are significantly different with p = 0.01)." how was this calculated, and given the many, many uncertainties that go into such a flux number one cannot possibly believe that this these numbers are indeed statistically significant.

Line 340: What about acidification induced changes linked to changes in the buffer capacity (see e.g. Hauck et al 2015)? A contribution within 17 years is plausible.

Line 439: I suppose ENSE is ENSO – but please either way spell out abbreviations when first using them

Line 505: "The fact that Chl-a is the dominant driver of interannual $\Delta$pCO2 variability should not be surprising" – the authors have not proven that chlorophyll is the dominant driver. From a pool of selected variables, it showed the largest correlation – this can barely be called a "fact" in science.

Figures 5 and 6: Uncertainties need to be added. Without uncertainty I do not trust that the observed variability is significant.

Figures 7 and 8: Wind stress anomalies are interesting, but direction would be equally interesting and provide more evidence.

---

## Referee Comment (RC2) · Anonymous Referee #2 · 6 Nov 2017

The authors present here a lengthy paper in which they aim to provide evidence that the included machine learning products for pCO2 are an appropriate proxy for investigating interannual variability in pCO2 and carbon fluxes in the Southern Ocean. They use these products to analyze transitions between regimes and also investigate the drivers of the changes. However, the paper needs to be proofread prior to submitting as it has errors throughout which distracts from the science. The job of a review is not to be a technical reviewer but to analyze the science. Unfortunately it is difficult to follow the science with the errors included throughout the paper.

The authors utilize an ensemble of five products for this study, however 2 of the products are just a repetition of two other products produced at a higher resolution. There is no evidence given that this results in 5 independent products for

this ensemble. At the very least, more discussion is required for it to be accepted that the high and low resolution versions of the same product can be seen as individual ensemble members. Additionally, the analysis should be revised to include the SOM-FFN created with SOCATv3 so that all members are produced using the same dataset. The SOM-FFN product is available based on SOCATv4 now and the authors should at least update it to the version based on SOCATv3 (see https://www.nodc.noaa.gov/ocads/oceans/SPCO2_1982_2015_ETH_SOM_FFN.html).

This paper is wrought with inconsistencies, incomplete definitions, and missed words which distracts the reader throughout. Some examples include the use of MIZ in text but AZ in figures, along with a consistent use of acronyms without first defining them (PFZ, AZ, MIZ, etc). Additionally, using the Fay & McKinley biome boundaries but then referring to the regions as SAZ, PFZ, AZ/MIZ is confusing and inaccurate. The biomes and frontal regions are not interchangeable and the authors need to be consistent throughout the study as to which they are using.

In Figure 2, the different extents into the ice covered regions of each product will affect the comparison shown in 2c. Ensure that equal regions are being compared. Also, showing how the products compare to the available data in Figure 2 would be helpful. Figure 3b is not discussed in the text at all. The definition of the "signal" as the largest difference in trend for a particular gridcell should be referenced as typically it is the mean trend/value that is the "signal" and the noise is the spread around that signal (either standard deviation, standard error, etc). Figures 5 and 6 need to be improved dramatically. The difference between the dark and light curves in Figure 5 is not defined. Additionally, there is no indication of which product (or average of the ensemble) is being plotted here. The captions are wrong (referenced subplots j-r which do not exist). The background gray shading to designate the difference regimes/timescales is difficult to distinguish. Perhaps another method to highlight those would be helpful. Additionally, trends included on these figures should have confidence intervals or uncertainty values included. Figures 7-8 continue with captions that do not correspond to

the figure ("wind stress (d-e), SST (f-h)" when wind stress is in (d-f) and SST in (g-i), etc.

Below are some obvious technical corrections which I have not already addressed above. I reiterate though that the entire paper needs to be revised and improved, as it is difficult to follow the science with incorrect references to figures and missing words in sentences.

Line 43: "not due to changes in overturning" Line 46: "The led to an oceanic dipole..." Line 62: repeated word Line 141: PFZ and MIZ not defined. MIZ and AZ used interchangeably throughout paper Line 149: missing a word in "the variability of between the ensemble members..." Line 157: how are "scores" calculated? Line 199: Should reference Figure 3c I believe Line 213: Figures 2a-c and 4a-c are not the correct reference figures for the points being addressed. Line 221: "CO2" Line 255-260: Figure 5 caption is inaccurate to what is shown in Figure 5 Line 289: Should reference Figure 5a,d rather than Figure 5 b,e. Line 291: remain consistent with capitalizing Southern Ocean Line 307: Do you mean the seasonal cycle amplitude? Line 342: "it was advanced that the explanation..." is awkward Line 350: "These studies have in linked the wind stress variability..." is awkward and needs to be rewritten. Lines 373-376: caption for Figure 7 and Figure 8 need to be corrected to accurately reference the figures. Line 387-388: "...and surrounds (Figure 7d,j)" is awkward. Consider revising sentence Line 397: "(-ve shift)". Is this supposed to be negative shift? If so, simply spell out to improve clarity. Line 439: "ENSE" perhaps should be ENSO?

Overall, the science presented in the paper is interesting and could provide a interested look at using various machine learning methods to gain an understanding of the Southern Ocean carbon drivers. However, the lack of proofreading prior to submitting is clear and must be improved before a complete review of the manuscript can be undertaken.

---

## Author Comment (AC1) · 27 Dec 2017

**Rebuttal to "Interannual drivers of the seasonal cycle of $CO_2$ fluxes in the Southern Ocean"**

*by* Gregor, Kok and Monteiro

**Overview of changes**

The authors would like to thank the reviewers for their recommendations to improve the manuscript. The changes are relatively substantial with the manuscript now being more succinct. The major scientific changes are: 1) removal of the two high resolution methods; 2) inclusion of uncertainty estimates. We also apologise for the multitude of typos and figure reference errors throughout the document. We hope that the reviewers now find the manuscript in a more presentable state.

In the document below we show the initial response of the reviewers in blue and the responses to each point in black. Track changes for the manuscript are shown below the response to the reviewer.

**Reviewer 1**

**Overview**

Gregor and co-authors investigate the variability of the Southern Ocean CO2 uptake strength from 1998-2014 analysed for 9 regions (SO divided in basins and Fay and McKinley biomes). The authors combine 5 realizations to form a multi-model mean which is used to investigate seasonality and year-to-year variability of the delta pCO2 and the CO2 flux. 3 of the realizations are independent whereas 2 are simply higher resolution versions of 2 other methods. The authors find that the seasonal variability is the strongest mode of variability in the SO. Additionally, the authors confirm results from past studies that the SO was losing some of its uptake capacity in the early years of their analysis period, then the uptake increased in the subsequent period, whereas in the final years of their analysis period the reinvigoration of the sink stopped again. The authors investigate the cause of this variability in the sink strength by analysing anomaly periods for winter and summer season separately. From this analysis the authors conclude that the drivers are seasonally decoupled with wind being the dominant driver for winter variability and biology being the dominant driver for summer variability.

I found this study to be interesting, comprehensive and in general suitable for the journal. The authors do not only present results from new methods to confirm previous results, they also deepen the analysis by looking at anomalies rather than trends (as previously done) and investigate different seasons.

**Major Issues**

- The study confuses trends and variability. When do the authors talk about trends and when about variability? At the moment, these two terms are mixed up. E.g. take figure 5 all panels. There is clearly some year-to-year variability causing in some years more and in some less uptake but overarching in all panels of figure 5 and 6 one can clearly see an increasing CO2 sink from 1998 onwards, i.e. an increasing trend throughout the entire time period. Furthermore, wording used like "decadal trends" and "interannual trends" contribute to the confusion. What is a decadal trend? Is it the slope of a regression line when considering at least 10 years of data? What are interannual trends? The same only considering 3 years? In the latter case one cannot speak of a trend at all.

  We have removed most of the references to trends. We have also removed the reference to the decadal mode of variability. We do, however, make the connection between the longer modes of variability in winter and that this is likely linked to the decadal variability mentioned by Landschützer et al. (2016) linked to the SAM.

- Despite being able to do so, the authors do not add uncertainties. There are many sources of uncertainty ignored by the authors, e.g. the measurement uncertainty (which is however neglectably small), the extrapolation error of the method, the building of the

multi-model mean creates an error and finally the calculation of the air-sea flux adds another source of uncertainty (through wind and transfer velocity choice). At the moment, the results are presented overconfidently. It is not clear how much of the explored variability is significant and how much is simply statistical gibberish. I am aware that there is no "perfect" way to represent all uncertainties, but in a data sparse region like the Southern Ocean a study like this needs to add the best possible uncertainty estimate, otherwise, many of the conclusions drawn cannot simply be accepted.

Regarding the propagation of errors, we used the same approach used by Landschützer et al. (2014). However, we do note that this error is Gaussian and mechanistically consistent, thus we can make deductions about the changes in the trends we observe. We also incorporate the between-method error from transCom (as in Gurney et al., 2004) at the recommendation of Christian Rödenbeck (pers comm.). We treat this, and a variation thereof as the primary threshold of significance as in Figures 5 and 6 (new numbering).

- On page 11 line 312 I found that the authors claim statistical significance between the mean uptakes, reporting a p-value. It is not clear to the reader what test was used and how significance has been determined. Also, when adding uncertainty, the authors will notice that an uptake of -0.17 PgC/yr is unlikely to be identified as statistically significantly different from -0.19 PgC/yr in the data sparse Southern Ocean.

We have removed the trends from the FCO2 time-series, so this is no longer an issue. Note that this figure has also moved to the supplementary materials.

- Despite the uncertainty of the CO2 itself there are other sources of concern related to uncertainty. Chlorophyll is e.g. also presented without discussing uncertainty. How is cloud coverage and missing chlorophyll data effecting the results? Also, wind products have been shown to have different trends locally in the Southern Ocean. This has not been mentioned. I am also wondering to what extend the use of different products hampers the conclusions of the manuscript. SST from Reynolds is based – to the extent of my knowledge – from satellite and in-situ data, whereas ECCO MLD is from an assimilation model. I would expect some disagreement between these products that have nothing to do with "real world" disagreement. This is not a massive concern, but certainly needs to be mentioned as well.

This is a valid point and we agree with the reviewers concern. However, it may be beyond the scope of this study to address the uncertainties driven by the input proxies. This may be a good topic for a collaborative effort of the SOCOM intercomparison. I have to some extent addressed the reviewers points:

**Chlorophyll:** we include the methodology used by Gregor et al. (2017) for the missing chlorophyll, where cloud patches are filled with the climatology for that point, and missing winter values are filled with a low level Gaussian noise.

**ECCO2 MLD:** This is indeed assimilation model output. Our motivation here is that

using observations (most likely the de Boyer Montégut (2004) MLD climatology) would mean that MLD could not treated as a potential driver of pCO2. However, there may be regions (such as the MIZ) where the MLD estimates are spurious for ECCO2. Choosing between these two products is thus a choice of trade-offs.

**Wind:** We include our justification for using CCMP v2 only. Much of this is based on the study by Swart et al. (2015a) and personal communication with Neil Swart.

- Another source of concern is the length choice of periods P1-P4. Periods P1-P3 are all of the same length, whereas P4 is substantially shorter. The authors claim that substantial year-to-year variability occurs on various timescales (4-6 year and 10 year modes respectively), hence the variation of periods aliases this analysis.

  We have changed the lengths of the periods to be more or less the same length (5, 4, 4, 4 years respectively). We hope that this approach addresses the reviewers concerns.

All the above points raised are of major concern and must be addressed before the manuscript can be considered for publication.

Minor comments:
- Title: the title only mentions CO2 fluxes whereas in the manuscript largely discusses delta pCO2

  We have made the title more generic: CO2
- Line 29: "paucity of observations (Landschützer et al 2015)" – This is not a good reference. Bakker et al 2016 (the SOCATv2 reference) would be a better reference for such a paucity.

  We have changed this reference to Bakker et al. (2016)
- Line 62: remove 2nd that

  removed
- Line 95: wrong reference (Bakker et al 2016)

  This reference is in fact correct for SOCAT v3. SOCAT v2 is Bakker et al. (2014)
- Table 1: How has the RMSE for the 0.25x0.25 degree product been calculated? SO-CAT offers a gridded product but on 1x1 degrees. Did the authors grid the 0.25 product themselves? If yes, has this been done the "SOCAT" way, i.e per cruise weighted or differently? Same with the 16-day time step. More information is needed here.

  We have removed the high resolution implementations of RFR and SVR at the request of Reviewer 2 (who felt they resulted in an ensemble biased towards these methods).
- Line 130: How different would the results be if a different transfer velocity was chosen? This may have significant effects on the uncertainty.

  The fluxes are now only included in the supplementary materials.
- Line 134: where are [ice] data from? I am missing a reference here.

*"...sea surface temperature (SST) and sea-ice fraction by Reynolds et al. (2007),…"*

- Line 175-176: "attributed this difference to the clustering step used by the SOM-FFN that created large discrepancies in the Atlantic sector." I have not found any convincing evidence for this in any of the cited papers.

  The published version of Gregor et al. (2017) now contains a figure (A3) that the reviewer may find convincing.

- Line 236: Either the authors used the wrong wording or there is a misunderstanding, but I do not see where the authors find that the delta pCO2 is zonally asymmetric within each biome. It looks like the other way around: Figure 4 looks like there is a strong zonal symmetry (besides indeed in panel a).

  Included an additional clause that the gradient is during summer: *"Apparent also from Figure 3 is that, over and above the latitudinal gradient, $\Delta pCO_2$ is zonally asymmetric within each biome during summer (Figure 3a), when biological uptake of $CO_2$ increases."*

- Lines 249-254: This paragraph is not clear. Please rephrase

  We have tried to make this more clear: *"The projected summer minima (dashed lines) are calculated by subtracting the mean seasonal amplitude from the winter maxima (Figure 4, with air-sea $CO_2$ fluxes shown in Figure S3). The projected summer minima is the expected summer $\Delta pCO_2$ under the assumption that summer $\Delta pCO_2$ is dependent on, but not restricted to, the baseline set by winter. Differences between the summer minima and projected minima are highlighted with green and blue patches, highlighting periods of decoupling between summer and winter interannual variability. The green areas indicate periods of strong uptake (relative to winter) that enhance the mean uptake of $CO_2$ and amplify the seasonal cycle. Conversely, blue areas show periods where weak summer uptake (relative to winter) offsets winter outgassing, thus reducing the mean $\Delta pCO_2$ as well as supressing the amplitude of the seasonal cycle (Figure 4)."*

- Line 273: interannual trend – what is that? Interannual variability. A 3-year trend? A trend that changes sign every year?

  We have rephrase this to: *"A key feature of Figure 4 is that the mean interannual variability is the net effect of decoupled seasonal modes of variability for summer and winter."*

- Lines 275: decadal mode: How are you able to say this. You have 1998-2014 data , i.e. 17 years of data. How can you detect a decadal mode from such a short time series?

  We have removed this statement and other references to the decadal mode, unless it is associated with previous studies (Lovenduski et al., 2008; Landschützer et al., 2016).

- Line 305: "decadal trend" – what is this? A trend of at least 1 decade?

  Removed decadal trend and see above comment

- Lines 311-312: "-0.19 and -0.17 PgC yr-1 for the Atlantic and Pacific sectors respectively

(where the latter are significantly different with p = 0.01)." how was this calculated, and given the many, many uncertainties that go into such a flux number one cannot possibly believe that this these numbers are indeed statistically significant.

We have moved the fluxes to supplementary materials. We have also removed the trends reported on the figure as this may detract and confuse given that the rest of the study deals with anomalies rather than trends.

- Line 340: What about acidification induced changes linked to changes in the buffer capacity (see e.g. Hauck et al 2015)? A contribution within 17 years is plausible.

This is an interesting point, but would lengthen the study too much, and is thus better left for another study.

- Line 439: I suppose ENSE is ENSO – but please either way spell out abbreviations when first using them

ENS(E)O is only used once and thus written out in full

- Line 505: "The fact that Chl-a is the dominant driver of interannual ΔpCO2 variability should not be surprising" – the authors have not proven that chlorophyll is the dominant driver. From a pool of selected variables, it showed the largest correlation – this can barely be called a "fact" in science.

Rephrased: *"Our finding that Chl-a is the dominant driver of interannual $\Delta pCO_2$ variability should not be surprising given that models and observations support this notion (Hoppema et al., 1999; Bakker et al., 2008; Mahadevan et al., 2011; Wang et al., 2012; Hauck et al., 2013; 2015; Shetye et al., 2015)"*

- Figures 5 and 6: Uncertainties need to be added. Without uncertainty I do not trust that the observed variability is significant.

We have added uncertainty estimates of uncertainty as previously mentioned. We keep these relatively simple in Figure 4 (old 5) by using time averages of the between-method errors. We do however include uncertainty thresholds in the final anomaly analysis. We feel that this is perhaps a more pertinent place to show the uncertainties.

- Figures 7 and 8: Wind stress anomalies are interesting, but direction would be equally interesting and provide more evidence.

We include the wind direction for each of the anomaly transitions in the supplementary materials. We find that the wind anomalies do not increase the understanding of the changes in ΔpCO2 significantly to justify the addition in Figures 5, 6 (old 7 and 8).

[revised manuscript text omitted]

---

## Author Comment (AC2) · 27 Dec 2017

**Rebuttal to "Interannual drivers of the seasonal cycle of $CO_2$ fluxes in the Southern Ocean"**

*by* Gregor, Kok and Monteiro

**Overview of changes**

The authors would like to thank the reviewers for their recommendations to improve the manuscript. The changes are relatively substantial with the manuscript now being more succinct. The major scientific changes are: 1) removal of the two high resolution methods; 2) inclusion of uncertainty estimates. We also apologise for the multitude of typos and figure reference errors throughout the document. We hope that the reviewers now find the manuscript in a more presentable state.

In the document below we show the initial response of the reviewers in blue and the responses to each point in black. Track changes for the manuscript are shown below the response to the reviewer.

**Reviewer 2**

**Overview**

The authors present here a lengthy paper in which they aim to provide evidence that the included machine learning products for pCO2 are an appropriate proxy for investigating interannual variability in pCO2 and carbon fluxes in the Southern Ocean. They use these products to analyse transitions between regimes and also investigate the drivers of the changes. However, the paper needs to be proofread prior to submitting as it has errors throughout which distracts from the science. The job of a review is not to be a technical reviewer but to analyse the science. Unfortunately it is difficult to follow the science with the errors included throughout the paper.

**Major revision**

The authors utilize an ensemble of five products for this study, however 2 of the products are just a repetition of two other products produced at a higher resolution. There is no evidence given that this results in 5 independent products for this ensemble. At the very least, more discussion is required for it to be accepted that the high and low resolution versions of the same product can be seen as individual ensemble members.

We acknowledge that these two high resolution implementations may not have contributed results that were mechanistically different to the low resolution implementations. We have thus removed the high resolution data from the study. This means that the now reduced ensemble includes only three machine learning methods: SOM-FFN, SVR, RFR.

Additionally, the analysis should be revised to include the SOM-FFN created with SOCATv3 so that all members are produced using the same dataset. The SOM-FFN product is available based on SOCATv4 now and the authors should at least update it to the version based on SOCATv3 (see https://www.nodc.noaa.gov/ocads/oceans/SPCO2_1982_2015_ETH_SOM_FFN.html ).

We now use v2.2 of the SOM-FFN implementation which used SOCAT v4. The results have been updated accordingly.

This paper is wrought with inconsistencies, incomplete definitions, and missed words which distracts the reader throughout. Some examples include the use of MIZ in text but AZ in figures, along with a consistent use of acronyms without first defining them (PFZ, AZ, MIZ, etc). Additionally, using the Fay & McKinley biome boundaries but then referring to the regions as SAZ, PFZ, AZ/MIZ is confusing and inaccurate. The biomes and frontal regions are not interchangeable and the authors need to be consistent throughout the study as to which they are using.

These abbreviations based on the Fay and McKinley (2014) paper where the authors state: *"...Southern Ocean regions that we define as STSS, SPSS and ICE biomes. Respectively, these three Southern Ocean biomes are comparable to the Sub-Antarctic Zone (SAZ), the Polar Frontal Zone (PFZ) and Antarctic Zone (AZ)"*. We chose to use MIZ instead of AZ. These biomes are now defined before they are used. If the reviewer still feels that these should be changed, we would be happy to make these changes.

In Figure 2, the different extents into the ice covered regions of each product will affect the comparison shown in 2c. Ensure that equal regions are being compared. Also, showing how the products compare to the available data in Figure 2 would be helpful.

We have changed the comparison area to be consistent for all models. This means that a large

portion of the MIZ/ICE is omitted. We have thus decided to exclude the MIZ from the rest of the analysis. We do show the MIZ data for both $\Delta pCO_2$ and $FCO_2$ in the Supplementary materials. The available data is unfortunately just too sparse to plot alongside the empirical estimates on a monthly scale as shown in the figure below.

[Figure]

**R1: Figure is the same as Figure 2 in the manuscript with the addition of the SOCAT v5 data.**

Figure 3b is not discussed in the text at all. The definition of the "signal" as the largest difference in trend for a particular grid cell should be referenced as typically it is the mean trend/value that is the "signal" and the noise is the spread around that signal (either standard deviation, standard error, etc).

Figure 3 has been removed from the manuscript. We now take a different approach in addressing the uncertainties at the request of Reviewer 1.

Figures 5 and 6 need to be improved dramatically. The difference between the dark and light curves in Figure 5 is not defined. Additionally, there is no indication of which product (or average of the ensemble) is being plotted here. The captions are wrong (referenced subplots j-r which do not exist). The background grey shading to designate the difference regimes/timescales is difficult to distinguish. Perhaps another method to highlight those would be helpful. Additionally, trends included on these figures should have confidence intervals or uncertainty values included.

We have changed this figure substantially. The MIZ has been dropped as the small remaining area (after masking for model inconsistency) is not representative of the biome. This has thus simplified the image a lot. We have darkened the shading as we too noticed that these are very light when the manuscript is printed. We have also included the dates of these periods on the figure. Note that the dates of the periods have changed at the request of Reviewer 1. The captions have been corrected.

We have also moved the figure showing the fluxes to the supplementary materials. We feel that this distracts slightly from the story that we are trying to tell. Moreover, we have removed the trends from the Flux figure as it is confusing to work with anomalies and then present trends (Reviewer 1).

Figures 7-8 continue with captions that do not correspond to the figure ("wind stress (d-e), SST (f-h)" when wind stress is in (d-f) and SST in (g-i), etc.
We have corrected these captions.

Technical Corrections
Below are some obvious technical corrections which I have not already addressed above. I reiterate though that the entire paper needs to be revised and improved, as it is difficult to follow the science with incorrect references to figures and missing words in sentences.

- Line 43: "not due to changes in overturning"
  I wasn't completely sure what the reviewer found at fault in this sentence other than being clumsy. It has been rephrased: *"While previous studies suggested that changes in wind strength have led to changes in meridional overturning and thus $CO_2$ uptake (Lenton and Matear, 2007; Lovenduski et al., 2007; Lenton et al., 2009; DeVries et al., 2017), Landschützer et al. (2015) suggested that atmospheric circulation has become more zonally asymmetric since the mid 2000's, which has led to an oceanic dipole of cooling and warming."*
- Line 46: "The led to an oceanic dipole. . ."
  *"...which has led to an oceanic dipole..."*
- Line 62: repeated word
  Removed
- Line 141: PFZ and MIZ not defined. MIZ and AZ used inter- changeably throughout paper
  We now define these when first introduced on L165.
- Line 149: missing a word in "the variability of between the ensemble members. . ."
  This has been rephrased to: *"The first section examines the uncertainties of the ensemble and its members."*

- Line 157: how are "scores" calculated?
  We now define the scores explicitly as the root mean squared error (RMSE).
- Line 199: Should reference Figure 3c I believe
  Figure has been removed
- Line 213: Figures 2a-c and 4a-c are not the correct reference figures for the points being addressed.
  The references to figures have (hopefully) been corrected
- Line 221: "CO2"
  We have corrected all occurrences of $CO_2$ to be subscripted
- Line 255-260: Figure 5 caption is inaccurate to what is shown in Figure 5
  Corrected and figure numbers have changed
- Line 289: Should reference Figure 5a,d rather than Figure 5 b,e.
  Corrected and figure numbers have changed
- Line 291: remain consistent with capitalizing Southern Ocean
  All occurrences of Southern Ocean have now been capitalised
- Line 307: Do you mean the seasonal cycle amplitude?
  We define this at the first occurrence of the phrase on L211
- Line 342: "it was advanced that the explanation. . ." is awkward
  This section has changed a great deal – this paragraph has been removed
- Line 350: "These studies have in linked the wind stress variability. . ." is awkward and needs to be rewritten.
  This has been removed
- Lines 373-376: caption for Figure 7 and Figure 8 need to be corrected to accurately reference the figures.
  The captions have been corrected
- Line 387-388: ". . .and surrounds (Figure 7d,j)" is awkward. Consider revising sentence
  Rephrased to: *This regional sustained saturation corresponds to a shift towards stronger winds and/or deeper MLDs in the west Pacific sector of the SAZ (Figure 5d,j).*
- Line 397: "(-ve shift)". Is this supposed to be negative shift? If so, simply spell out to improve clarity.
  All occurrences of +ve and –ve have been written out as positive and negative respectively.
- Line 439: "ENSE" perhaps should be ENSO?
  Changed to ENSO

Overall, the science presented in the paper is interesting and could provide a interested look at using various machine learning methods to gain an understanding of the Southern Ocean carbon drivers. However, the lack of proofreading prior to submitting is clear and must be improved before a complete review of the manuscript can be undertaken.

[revised manuscript text omitted]

---

## Referee Report (RR1)

Reference for Interannual driver of the seasonal cycle of CO2 fluxes in the Southern Ocean
Luke Gregor, Schalk Kok, and Pedro M. S. Monteiro

This paper is much improved and I thank the authors for considering each of the concerns and recommendations of the first-round reviewers. I think with some minor revisions, this paper will be ready for publication and an excellent contribution to the field. Below are a few comments and suggestions. Also, the included line numbers seem to not extend past 100 before starting again from 00 so I include page and line numbers in order to help locate my references.

I think that Figure 2 is very clear to help the reader visualize the products described here. Figure 2d is specifically interesting. However, it should be considered that the amount of area covered by the summer MIZ region is different from the winter MIZ region. When calculating the standard deviation you need to account for that.

Figure 3: The ice mask varies by season in this figure but I don't understand what the source of the mask is. The SOM-FFN product specifically has the same coverage through all seasons I know. Is the mask just the regions where all 3 ensemble members have values for that season? I could see that the chosen MLD or Chl product input could limit this coverage during certain seasons, but just making it clear where that comes from would be helpful. Also, you could consider not showing the MIZ region all together since you state on Page 9, in line 21 that you are excluding it from the paper.

Throughout the manuscript, I strongly suggest you take care when using the word "data" to describe the output form these machine learning methods. Someone not as familiar with the topic could be led to believe we actually have observations everywhere that you show (for example on Page 11, line 84).

Lastly, in the synthesis, it should be noted that this shorter timeframe could bias/limit the results presented here and only with increased timeseries of not only $pCO_2$ but also these drivers (and the need for continued sustained satellite observations) will this work be validated and improved upon.

---

## Author Response (AR2)

**Reply to the reviewers for the second review of Gregor et al: Interannual drivers of the seasonal cycle of CO2 in the Southern Ocean**

We would like to thank the reviewers for their thorough reviews of the paper and we appreciate the time taken to hone the manuscript towards publication. We have made relatively large changes to the manuscript with changes to both figures and text. In particular we have addressed the issue of uncertainty a lot more thoroughly. This means that the content of section 3.5 (winter) has changed a fair deal. In addition we have also made use of a professional editor, which has hopefully reduced the amount of editorial mistakes in the manuscript to an acceptable amount.

Below we address the comments of the reviewers in blue. After that we show the track changes for the document.

**2nd review of Gregor et al: Interannual drivers of the seasonal cycle of CO2 in the Southern Ocean**

Response to previous concerns:

Previously, I have raised 3 main concerns. The first related to the very confusing presentation of variability and trends, the second related to the missing evaluation of uncertainties (of various sources) and the third related to the choice of time periods as they are compared in the text.

In their revised manuscript, the authors have indeed taken into consideration the concerns raised by this reviewer, however, as I will outline below, they still miss to clearly communicate the uncertainty of their analysis and thereby still present results overconfidently.

Firstly though, the authors have done a good job in their manuscript to clarify when they consider trends and variability and what is the time period considered. This makes the manuscript much easier to read. Exceptions however still exist, e.g. the abstract line 21 (here authors talk about "interannual trends") and on page 21 line 70, where the authors state "... summer drivers may explain the inter-annual variability in the decadal trends". These statements need clarification – see my previous review.

We have changed the wording to avoid the confusing the decadal trends (with this short timeseries); however, in the first part of the paragraph on page 21, we refer to the decadal trends in the cited publications, which do consider longer time series from which decadal variability can be derived.

Coming back to the uncertainty analysis: The authors now provide an error assessment which is a great leap forward, however, when discussing the results the new uncertainty estimates are often mentioned but likewise not properly taken into consideration. To give you a concrete example: Uncertainties are still only hand-wavy included in figure 4 which makes me doubt that these observed short temporal variabilities are real or just statistical noise given that these fluctuations are often in the order of 2-3  $\mu$ atm (visually assessment based on figure 4). Figure 2 likewise suggests errors beyond the displayed differences between methods. Another example is figures 5 and 6. Here the authors do add the uncertainty, but fail to properly discuss the limitation that basically the majority of the SO variability is insignificant, besides a few regions. Instead the authors assume the significant regional drivers are representative for the entire region. The authors further only mention uncertainty in a first sentence of the sections but then it is not clear if this is properly taken into consideration when discussing the drivers (see specific comments below).

We have changed figure 4 to now show the uncertainty surrounding ensemble estimates for summer and winter. We feel that this is now far more explicit around the error. Moreover, we realise that the interannual variability is

Regarding my third point, the authors still don't make a strong enough case for their periods they consider. They refer in the text to figure 4 but visually it is not obvious why the periods have been chosen. I am aware that this is a new argument (as I have previously only criticized the length of the periods). This however can be easily solved by adding a sentence or 2 explaining why these periods were chosen (may this be due to some metric or subjective choice)

We have changed the motivation for our time periods slightly. We also add a sentence at the

end of the paragraph that makes it clear that we do not feel that our study is dependent on these specific period lengths or starting years: "In order to capture the decoupled 4–6 year short-term variability observed in summer, the estimates are divided into four objectively selected periods (P1 to P4). The periods are each four years long with the exception of P1, which is five years long due to the fact that the time-series is not divisible by four (with a length of 17 years in total). Given a longer time series, this analysis would benefit from testing different lengths for each period, as well as varying the starting and end years."

In summary, I think the manuscript has improved, but overall, issues remain regarding the uncertainty of the analysis. This really pains me because I do think that the paper is important and I do very much like the approach based on looking at anomalous periods (rather than linear trends). Also, an assessment based on 2 novel methods is a welcome addition to assessment papers such as the recently published Ritter et al. (GRL) Southern Ocean SOCOM trend comparison. I don't think that (many of) the conclusions drawn would fall based on the error assessment, but in a data sparse region like the Southern Ocean, where all methods rely on heavy data extrapolation the uncertainty must be on the forefront of any variability, trend or process study.

**Recommendation:**

Based on the revision, I cannot recommend the manuscript for publication. Instead I would like to see a revised manuscript, where the authors really discuss their results in in a fair way in light of the uncertainty they are facing. Plus I suggest the authors check remaining editorial issues. I am convinced that after this step (plus some minor comments below) the paper can become acceptable for publication in BG.

**Specific comments:**

In general: Many editorial issues need to be fixed. E.g. in many instances commas are missing, the authors switch between present tense and past tense (e.g. in the abstract) and figure 7 is labelled figure 9. I will not list them all here, but rather suggest professional text editing.

We have sent the document to a professional editor and we hope that the mistakes have now been reduced to an acceptable amount.

Abstract line 9-10: (a) very minor but SOCAT includes fCO2 not pCO2 and (b) "... ship measurements of pCO2 (SOCAT) ..." really is a clumsy way to introduce SOCAT. Firstly, the abbreviation SOCAT needs to be defined (Surface Ocean CO2 Atlas) and secondly, what about LDEO? This database equally includes pCO2 ship measurements. We leave the introduction of SOCAT to the methods section of the manuscript.

Abstract line 13: "... nine regions defined by basin ..." – at this time you have not mentioned the Southern Ocean so the reader gets the impression you talk about the actual basins. We have made this more clear; however, it should be clear from the title that we consider the Southern Ocean in this study.

Abstract line 15: delta pCO2 is not defined (i.e that you mean the difference between ocean and atmosphere – it may as well be the seasonal difference). Changed to accommodate the seasonal differences

Abstract line 21: "Interannual trends" - see my previous assessment This has been addressed: we refer now only to interannual variability.

Abstract line 22: "... chlorophyll-a variability where the latter had high mean seasonal concentrations." It is not clear what the authors try to say here We have tried to clarify this: where higher concentrations of chlorophyll-*a* correspond with lower  $pCO_2$  concentrations.

Introduction line 32: "accurately measure" – I suggest "accurately quantify". Measurements of any quantity have reached high accuracy. The interpretation through interpolation methods (such as this study – hence the necessity of an uncertainty estimate) suffer from lower accuracy.

Changed as suggested

Introduction lines 34 and 35: "Empirical models provide an interim solution to this challenge until prognostic ocean biogeochemical models are able to represent the Southern Ocean CO2 seasonal 35 cycle accurately" – it is not clear from the context of the text why the seasonality is suddenly important here

Have made this a little more general for the first paragraph, we then later address the importance of the seasonal cycle later in the introduction.

Introduction line 37: "source in the 1990's" – I am not aware of any study that suggests the Southern Ocean was a source in the 1990's. Studies of Le Quéré and Lovenduski only suggest a saturation of the sink. Do you refer to a specific region or a specific season or both? Changed to weakening sink / strengthening sink

Introduction line 44: Not all proxies in the literature are satellite proxies. Have changed this: *The proxies are often satellite observable, but may include climatologies or output from assimilative models.*

Introduction line 47: The Landschutzer et al 2015 paper focuses on 2002-2011 and not 2000-2010

Changed accordingly

Introduction line 56-57: Additionally, the Xue et al 2015 paper suggests the same trends based on observations south of Tasmania and should be cited.

Included the Xue et al. (2015) study: *This is supported by observations from the Drake Passage and south of Australia showing that variability of upwelling has affected*  $\Delta pCO_2$ *(Munro et al., 2015; Xue et al., 2015).* We discuss the role of the SAM in driving this variability in the following paragraph, where Xue et al. (2015) is cited again.

Page 3 line 92: It is "Self-Organizing Map" – i.e. singular not plural. Corrected

Page 4 line 2: "v2.2" This is not the SOM-FFN version the authors refer to, but the version of the database where the data are stored. Changed this identifier to the run id: netG05

Results: See also main comments above.

Page 4 line 18: "comparing the different products is beyond the scope of this study" is a clumsy formulation. The authors do compare products here, but pCO2. The phrase should rather read that comparing proxies is beyond the scope of this study ... comparing the different proxies used in each of the CO2 products is beyond the scope of this study.

Page 6 line 73: The first sentence is not necessary – of course you discuss the results in the results section Sentence removed

Figures 2 and 4: Add uncertainty alongside the lines. Not as numbers. It is difficult to compare lines with numbers. At the moment, it looks like the authors try and highlight a difference between methods in Figure 2 that is not statistically significant (given the Ew and Eb numbers) as well as amplitude anomalies (green and blue) in Figure 4 that are as well not significant based on the Eb. This is very confusing. So, my question to the authors is: Can you actually say – with absolute certainty – that (a) any of the 3 methods is at any given point in time statistically significantly different from any of the other methods? (b) That anomalies are - with absolute certainty - the result of environmental conditions and not simply the result of internal variability? Based on the evidence presented, I doubt you can. We are now far more cautious about our results. We also show the uncertainties in Figure 4. Figure 5 is dedicated to showing the regional errors (moved from the supplementary materials), and we also show uncertainty in the plots for the drivers (Figures 6 and 7). Addressing now the specific points: (a) we agree that the methods are likely not statistically significant at any given time; however we do feel that it is necessary to explain the observed differences as they contribute to the between method error. (b) We are a bit more cautious with ascribing the drivers to changes in  $\Delta pCO2$  and describe the changes only where the between method uncertainty is acceptable.

Page 8 line 10: The authors missed my point in the first review round. I have noted that I have not seen any evidence that the CLUSTERING step is causing the difference. I am well aware that there is a difference and I do trust the authors with their assessment that the difference comes from the SOM method, but in neither of the papers I have seen any evidence that it is in fact the CLUSTERING step responsible for the mismatch. Many people are using the Landschutzer product, hence such an assessment of the cluster-based mismatch would be very valuable to the community. So, in summary: it is not enough to point at a difference plot and jumping to mechanistic conclusions. The authors should rather add a more in-depth analysis – also comparing the products to actual observations - if they want to add such a conclusive statement.

We've removed the statement that clustering is the driver of this difference, but may look into this in a bit more detail in a later study.

Page 9 line 21: Is the MIZ now in- or excluded? Later on, it is mentioned again. And if it is excluded, then why mention it at all?

Have removed panel c from Figure 2 (MIZ) and masked in Figure 3. References to the MIZ in the remainder of the text have been removed.

Page 9 line 34: Figure 3 a-d Corrected Page 10 line 54: Now the MIZ is discussed again – very confusing Removed this sentence

Page 10 line 65: Only 6 are shown? Why? Is the MIZ in or out? It seems that it is ignored in the figures but added to the text. This is misleading the reader. We have removed the MIZ estimates from the main text – inconsistent coverage between methods, large errors (stemming from little data).

Page 11 line 88: "however, our confidence in the changing trend is low due to lack of coherence between methods (Figure 2a,b) and only three years of data, with little data in 2014." – This statement is a bit of a surprise. Here the authors highlight that their trends are uncertain, but in the following they discuss the short term IAV as if it only little uncertainty is present. In contrast, Rodenbeck et al 2015, Landschutzer et al 2015 and Ritter et al 2017 show that trends are more robust among methods than IAV. Please explain or expand. We have now addressed the variability of IAV between methods more explicitly. We hope that this addresses the reviewers concerns. Moreover, we have removed this statement as we agree that it is conceptually inconsistent.

Page 13 line 35: "The 335 mean of the method anomalies for each transition is then taken. These anomalies are considered significant if the absolute estimate of the anomaly is larger than the standard deviation between the methods for each period" – all fine, but I am puzzled why one uncertainty estimate is in the methods section and the other is in the appendix? The section in the supplementary materials about the calculation of uncertainty in Figures 6 and 7 has been moved to the main text.

Page 13 line 47: The authors here mention that the uncertainty estimate masks out large regions. They equally and rightfully point out that there are other regions that are not masked and that those are considered. I do agree with the author's driver assessment in the following but now my question: Based on the assessment of the fewer, significant regions, how much can one assume that the driver assessment is also driving the variability of the larger – insignificant SO. I don't think one can with absolute certainty.

Plotted the uncertainty mask on the driver anomalies too. We have adjusted the text accordingly – this has led to substation changes in Section 3.5.

Page 15 line 99: "However, seasonal – regional analysis shows that the observed relationship between pCO2 and SST is counterintuitive (Figure 5a-c,g-i). On this basis we propose that SST is not a driver of pCO2 in winter." – Hold on here: Firstly, this is not a new proposal but has been e.g. shown by Takahashi et al 2002. Secondly, despite temperature not being the driver, the solubility relation still exists, it is simply not dominating the variability (see e.g. Figure 3 of Landschutzer 2015). Thirdly, 2-3 lines earlier the authors mention that they propose changes in wind stress as an alternative hypothesis to Landschutzer 2015, but this is exactly the point of the Landschutzer paper, that changes in the wind pattern and thereby changes in wind stress and upwelling caused the reinvigoration of the SO sink (see again e.g. Figure 3 in Landschutzer et al). The main difference is that these authors have not done the analysis for winter separately.

We have changed this paragraph accordingly to state that the finding is a refinement of the hypothesis put forward by Landschützer et al. (2015) in that we add a seasonal constraint.

Page 16 lines 28 onward: The authors talk about correlations, but based on the visual comparison it is not easy to verify this assessment. It would help to add an actual correlation plot, or adjust the colour scheme.

We have added the correlations for each driver anomaly with  $\Delta pCO2$  anomalies on the respective driver. This has been done for Figures 6 and 7

Page 16 line 35: "Looking more specifically at the significant variability..." – Now I am completely lost. Do the authors now, as they state in the beginning, only consider significant regions or not? This statement suggests that they did not but start doing so now. This is has been removed to avoid this confusion.

**Suggestions for revision or reasons for rejection (will be published if the paper is accepted for final publication)**

Reference for Interannual driver of the seasonal cycle of CO2 fluxes in the Southern Ocean Luke Gregor, Schalk Kok, and Pedro M. S. Monteiro

This paper is much improved and I thank the authors for considering each of the concerns and recommendations of the first-round reviewers. I think with some minor revisions, this paper will be ready for publication and an excellent contribution to the field. Below are a few comments and suggestions. Also, the included line numbers seem to not extend past 100 before starting again from 00 so I include page and line numbers in order to help locate my references.

I think that Figure 2 is very clear to help the reader visualize the products described here. Figure 2d is specifically interesting. However, it should be considered that the amount of area covered by the summer MIZ region is different from the winter MIZ region. When calculating the standard deviation you need to account for that.

We have removed the references to the MIZ at the recommendation of Reviewer 1. We show the plots of the MIZ in the Supplementary materials.

Figure 3: The ice mask varies by season in this figure but I don't understand what the source of the mask is. The SOM-FFN product specifically has the same coverage through all seasons I know. Is the mask just the regions where all 3 ensemble members have values for that season? I could see that the chosen MLD or Chl product input could limit this coverage during certain seasons, but just making it clear where that comes from would be helpful. Also, you could consider not showing the MIZ region all together since you state on Page 9, in line 21 that you are excluding it from the paper.

We have adopted the recommendation, masking the MIZ for Figure 3.

Throughout the manuscript, I strongly suggest you take care when using the word "data" to describe the output form these machine learning methods. Someone not as familiar with the topic could be led to believe we actually have observations everywhere that you show (for example on Page 11, line 84).

We have changed data (referring to interpolated data) to estimates.

Lastly, in the synthesis, it should be noted that this shorter timeframe could bias/limit the results presented here and only with increased timeseries of not only pCO2 but also these drivers (and the need for continued sustained satellite observations) will this work be validated and improved upon.

We have added this statement to the summary.

**Interannual drivers of the seasonal cycle of CO2 in the Southern**

**Ocean**

5

Luke Gregor1,2, Schalk Kok3 and Pedro M. S. Monteiro1

1 Southern Ocean Carbon-Climate Observatory (SOCCO), CSIR, Cape Town, South Africa

2 University of Cape Town, Department of Oceanography, Cape Town, South Africa

3 University of Pretoria, Department of Mechanical and Aeronautical Engineering, Pretoria, South Africa *Correspondence to*: Luke Gregor (luke.gregor@uct.ac.za)

**Abstract.**

- Resolving and understanding the drivers of variability of CO2 in the Southern Ocean and its potential 10 climate feedback is one of the major scientific challenges of the ocean-climate community. Here we use a regional approach on empirical estimates of  $pCO_2$  to understand the role that seasonal variability has on long term CO2 changes in the Southern Ocean. Machine learning has become a useful the preferred empirical modelling tool to interpolate time- and location-restricted ship measurements of pCO2 (SOCAT) to a gridded map using satellite data. In this study we use an ensemble of three machine--15 learning methods products: Support Vector Regression (SVR) and Random Forest Regression (RFR) from Gregor et al. (2017);), and the SOM-FFN method from Landschützer et al. (2016). The interpolated data were estimates of  $\Delta p CO_2$  are separated into nine regions in the Southern Ocean defined by basin (Indian, Pacific and Atlantic) and biomes (as defined by Fay and McKinley, 2014a). The regional approach showed a meridional gradient and zonal asymmetry in the magnitude of  $\Delta p CO_2$  estimates. Importantly, 20 there was ashows that, while there is good agreement in the overall trend of the products, there are periods and regions where the confidence in estimated  $\Delta p CO_2$  is low due to disagreement between the products. The regional breakdown of the data highlighted the seasonal decoupling of the modes for summer and
- winter interannual variability. Winter interannual variability had a longer mode of variability compared to summer, which varied on a 4–6 year time scale. To understand this variability of Δ*p*CO2, we
  investigated changes in summer and winter Δ*p*CO2 and the drivers thereof. The dominant winter changes are driven by wind stress variability. This is consistent with the temporal and spatial characteristics of the Southern Annular Mode (SAM), which has a decadal mode of variability (Lovenduski et al., 2008; Landschützer et al., 2016). Interannual trends in summer variability of Δ*p*CO2 are consistent with
- 30 of the  $\Delta pCO_2$  and its drivers into summer and winter. We find that understanding the variability of  $\Delta pCO_2$ and its drivers on shorter time scales is critical to resolving the long-term variability of  $\Delta pCO_2$ . Results show that  $\Delta pCO_2$  is rarely driven by thermodynamics during winter, but rather by mixing and stratification due to the stronger correlation of  $\Delta pCO_2$  variability with mixed layer depth. Summer  $pCO_2$

chlorophyll-a variability where the latter had high mean seasonal concentrations. We separate the analysis

variability is consistent with chlorophyll-a variability, where higher concentrations of chlorophyll-a 35 correspond with lower  $pCO_2$  concentrations. In regions of low chlorophyll-*a* concentrations, wind stress and sea surface temperature emerged as stronger drivers of  $\Delta p CO_2$ . In summary we propose that subdecadal variability is explained by summer drivers, while winter variability contributes to the long\_term changes associated with the SAM. This approach is a useful framework to assess the drivers of  $\Delta pCO_2$ but would greatly benefit from improved estimates of  $\Delta p CO_2$  and a longer time series.

**40 **1** Introduction**

The Southern Ocean plays a key role in the uptake of anthropogenic CO2 (Khatiwala et al., 2013; DeVries et al., 2017). Moreover, it has been shown that the Southern Ocean is sensitive to anthropogenically influenced climate variability, such as the intensification of the westerlies (Le Quéré et al., 2007; Lenton et al., 2009; Swart and Fyfe, 2012; DeVries et al., 2017). Until recently, the research community has not 45 been able to accurately measurequantify the contemporary changes, let alone understand the drivers, of CO2 in the Southern Ocean accurately due to a paucity of observations, let alone understand the drivers (Bakker et al., 2016). Empirical models provide an interim solution to this challenge until prognostic ocean biogeochemical models are able to represent the Southern Ocean CO2 seasonal cycle accuratelyfluxes adequately (Lenton et al., 2013; Rödenbeck et al., 2015; Mongwe et al., 2016). The 50 research community agrees on large changes in CO2 fluxes in the Southern Ocean from a source in the 1990's to a weakening sink in the 2000's 1990s to a strengthening sink in the 2000s; however, there is disagreement inover the drivers of the changes in CO2 uptake (Lovenduski et al., 2008; Landschützer et al., 2015; DeVries et al., 2017); Ritter et al., 2017). This study aims to understand the drivers of the changing CO2 sink in the Southern Ocean, based on an ensemble of empirical estimates using a seasonal analysis framework.

55

Empirical methods estimate CO2 by extrapolating the sparse ship-based CO2 measurements using proxy variables. The proxies are often satellite observable proxies. This approach has allowed for a better but may include climatologies or output from assimilative models. Empirical methods have improved our

- 60 understanding of the drivers of CO2 trends in the Southern Ocean by providing improved spatial and temporal resolution increasing the data coverage. However, there is still disagreement between many of the variability methods due to the paucity of data and the way in which each method interpolates sparse data (Rödenbeck et al., 2015; Ritter et al., 2017).
- 65 In a key study, Landschützer et al. (2015) showed, using an artificial neural network (ANN), that there was significant strengthening of Southern Ocean  $CO_2$  uptake during the period 20002002-2010. While previous studies suggested that changes in wind strength have led to changes in meridional overturning

and thus CO2 uptake (Lenton and Matear, 2007; Lovenduski et al., 2007; Lenton et al., 2009; DeVries et al., 2017), Landschützer et al. (2015) suggested that atmospheric circulation has become more zonally asymmetric since the mid 2000's2000s, which has led to an oceanic dipole of cooling and warming. The net impact of cooling and warming, together with changes in the DIC/TA (Dissolved Inorganic Carbon/Total Alkalinity), led to an increase in the uptake of CO2 (Landschützer et al., 2015). During this period, southward advection in the Atlantic basin, southward advection reduced upwelled DIC in surface waters, overcoming the effect of the concomitant warming in the region. Conversely, in the Eastern Pacific sector of the Southern Ocean, strong cooling overwhelmed increased upwelling (Landschützer et al., 2015). Munro et al. (2015)This is supported this mechanism, with databy observations from the Drake Passage and south of Australia showing that Avariability of upwelling has affected  $\Delta p$ CO2 decreased between 2002 and 2014.(Munro et al., 2015; Xue et al., 2015).

[revised manuscript text omitted]

|                | Propagated errors (µatm) |       |        | Within    | Between                 |
|----------------|--------------------------|-------|--------|-----------|-------------------------|
| Biome          | SVR                      | RFR   | SOM    | hod error | model me
thod |
|                |                          |       | SOM-   | (uatm)    | error                   |
|                |                          |       | I'I'IN | (1)       | (µatm)                  |
| SAZ            | 17.48                    | 14.50 | 12.30  | 14.91     | 4.88                    |
| PFZ            | 15.94                    | 12.71 | 13.09  | 13.99     | 4.78                    |
| MIZ            | 36.38                    | 24.53 | 22.46  | 28.46     | 10.81                   |
| Southern Ocean | 25.06                    | 17.91 | 16.44  | 20.16     | 6.79                    |

We use the RMSE scores as presented in Gregor et al. (2017) with abbreviated results shown in Table 1. The SOM-FFN method has the best score (14.84 μatm). SVR scores the lowest (24.04 μatm), but wasis still included due to the method's sensitivity to sparse data, which is favourable to the poorly sampled winter period (Gregor et al-2 2017). This compliments the RFR method, which scores well (16.45 μatm), but is prone to being insensitive to sparse data (Gregor et al-2 2017). These RMSE scores are used to calculate the total errors for each method and region using equation (3), where the measurement and mapping errors are both 5 μatm each (Pfeil et al., 2013; Sabine et al., 2013). These results are shown in Table 2.

215

20

Total errors are used to calculate the within-method error, which is an estimate of the combined total errorserror of the three machine-learning methods (equation 4). The between-method errors are the mean of the standard deviation between the methods (equation 5). The within-method errors are much larger than the between-method errors (Table 2). However, the within-method errors are normally distributed and are mechanistically consistent (Gregor et al., 2017). This allows us to observe changes that are smaller than the within method error. The between-method error (shown in Figure 2d) serves as a better measure of 2c) is thus used to determine whether observed variability is more than statistical noise as it incorporates consistent between the three methodologically different approaches. methods.